# Investigation of the demand for a 7-day (extended access) primary care service: an observational study from pilot schemes in England

William Whittaker,[1] Laura Anselmi,[1] Pauline Nelson,[2,3] Caroline O'Donnell,[3] Natalie Ross,[2] Katy Rothwell,[3] Damian Hodgson[2,3]

¹Manchester Centre for Health Economics, University of Manchester, Manchester, UK
²Alliance Manchester Business School, University of Manchester, Manchester, UK
³NIHR CLAHRC Greater Manchester, Manchester, UK

**Correspondence to**
Dr William Whittaker;
William.whittaker@manchester.ac.uk

## ABSTRACT

**Objectives** To understand how the uptake of an extended primary care service in the evenings and weekend varied by day of week and over time. Secondary objectives were to understand patient demographics of users of the service and how these varied by type of appointment and to core hour users.

**Design** Observational study.

**Setting** Primary care extended access appointments data in 13 centres in Greater Manchester, England, during 2016.

**Participants** Appointments could be booked by 1 261 326 patients registered with a family practitioner in five Clinical Commissioning Group geographic areas.

**Main outcome measures** Primary outcome measure was whether an appointment was used (booked and attended), secondary outcome measures included whether used appointments were prebooked or booked the same day, and delivered by a family or nurse practitioner. Additional analyses compared patient demographics with patients reporting the use of core hour primary care services.

**Results** 65.33% of 42 472 appointments were booked and attended (used). Usage of appointments was lowest on a Sunday at 46.73% (18.07 percentage points lower usage than on Mondays (95% CI −32.46 to −3.68)). Prebooked appointments were less likely to be booked among age group 0–9 and to result in patients not attending an appointment. Family practitioner appointments were increasingly less likely to be booked with age in comparison to nurse appointments. Patients attending extended access appointments tended to be younger in comparison to core hour patients.

**Conclusions** There is spare capacity in the extended access service, particularly on Sundays, suggesting reconfigurations of the service may be needed to improve efficiency of delivering the service. Patient demographics suggest the service is used by a relatively younger population than core hour services. Patient demographics varied with the types of appointment provided, these findings may help healthcare providers improve usage by tailoring appointment provision to local populations.

## Strengths and limitations of this study

► The first study assessing same-day and prebookable appointments at evenings and weekends in general practice.
► The first study comparing demographics of patients using evening and weekend appointments to patients using core hour appointments.
► The first study assessing uptake of evening and weekend appointments over time.
► The study covers five areas with differing provision, but may not be generalisable to other settings.
► No information on child use of core hours was available, limiting an assessment of this group of the population.

report difficulties in accessing primary care, particularly outside of core working hours.[2 3] Constrained access to primary care has been a reason for increased patient dissatisfaction and use of emergency care in a variety of countries and settings.[4] Rising use of emergency department services and patient dissatisfaction with access have highlighted a need for better access to primary care.

In England, primary care core hour services are provided by general practitioners (GP) from 08:00 to 18:30 Monday to Friday either by prebooking or calling on the same day. The National Health Service (NHS) in England aims to extend access to primary care services in the evening and at weekends as part of their strategy for delivering primary care by 2020/2021.[5] The strategy seeks to enable local commissioners of healthcare to redesign primary care services and commission extra capacity so that by 2020 '*everyone has access to GP services, including sufficient routine appointments at evenings and weekends to meet locally determined demand, alongside effective access to out-of-hours (OoH) and urgent care services.*' Extended access has been

## INTRODUCTION

Providing access to healthcare services when needed is a key target for health systems worldwide.[1] However, patients commonly

---

| **Box 1   Description of Greater Manchester extended access appointments in primary care** |
| :--- |
| **What is the service?**<br>Extended access appointments are appointments in primary care outside of core hours (typically from 08:00 to 18:30). Appointments are provided in addition to usual care outside of core family practitioner hours which include: walk-in centres, out-of-hour services, accident and emergency/secondary care, NHS Direct ('111'). The services may have been designed to replace existing Direct Enhanced Services arrangements (additional hours of opening contractually agreed with NHS England). |
| **Who provided the service?**<br>Extended access appointments are appointments with either a GP or nurse practitioner. |
| **How is the service delivered?**<br>Extended access appointments are typically face-to-face appointments delivered individually. |
| **Where is the service delivered?**<br>Extended access appointments are usually provided through one practice or similar premises which act as a local 'hub' covering several practices in a given geographic area, with between one and four hubs serving the patient population. Hub locations were decided by local healthcare commissioners (Clinical Commissioning Groups, CCG). CCGs are responsible for the commissioning of healthcare to a local, geographically defined population comprising a number of family practitioner (GP) practices. |
| **When and how much of the service is delivered?**<br>Extended access appointments under this study became operational at varying times over the 2016 calendar year, this varied by CCG. |
| **Tailoring of the service**<br>The type, referral route(s) and timing of appointments are decided by the local CCG. Variations between schemes existed by referral route, whether prebookable appointments were available, discipline of appointment (GP or nurse, all face to face), time and day of appointments. |

piloted nationally by NHS England since 2013 through the Prime Minister's Challenge Fund (PMCF), and its successor the GP Access Fund (GPAF). Wave 1 covered 7.5 million people in 2014, rising to 18 million (a third of the population of England) by 2015 in Wave 2. In parallel, NHS England Greater Manchester, now the Greater Manchester Health and Social Care Partnership (GMHSCP), has also piloted extended access appointments with an initial wave in 2014,[6] followed by a roll-out throughout Greater Manchester (GM) from 2016 in line with the region's devolution and health and social care strategy.[7] A description of extended access appointments in GM is provided in box 1.

Extended access evaluations to date have found reductions in emergency service use for patients with minor injuries/issues in national,[8] subnational[9] and GM[6 10] schemes. However, the reductions in emergency service use alone may not sufficiently offset the cost of providing an extended service, raising concerns that an extended access service is not a cost-effective use of health system resources.[10] Concerns of opportunity cost (the net

benefits of services that could have been provided instead of extended access) are particularly pertinent in times of increasing financial/workforce pressures for many health systems. However, in addition to impacts on secondary care, and particularly in the current absence of an understanding of the long-term health benefits of the service, there are additional aspects of extended access that need to be considered when assessing the value of the service. First, there is a lack of knowledge concerning how the service is being used with little assessment of how best to operationalise extended appointments efficiently (which would have implications for cost-effectiveness). One way to measure efficiency could be uptake of the service, with unbooked appointments signalling inefficient use of resources. Uptake of extended access appointments has been reported as 71% with lower uptake on weekends, particularly Sundays.[8 11] A subnational evaluation of three schemes found uptake was lower where appointments were not prebookable, and lower on Sundays.[11] An evaluation of the initial wave of GM 'demonstrators' found uptake of 65.5% and lower uptake on weekends, particularly on Sundays.[6] Studies of uptake have concentrated on high-level aggregated descriptive statistics of uptake over time and/or by day of week. Since service provision varies across pilots it is also important to understand how uptake varies across services delivered. Commissioners are tasked with delivering extended access services in light of patient demand (5, p 48), the lack of evidence to inform efficient provision is a substantial gap in knowledge. Second, it is currently unknown what the patient demographics are for an extended access service delivering prebookable and same-day appointments with existing analyses being for small-scale schemes[11] or urgent care appointments only,[12] it is also unclear whether patients using the service differ from patients using core hours general practice, understanding these may help inform service provision and infer whether the service is hitting an unmet need and/or reducing pressures in core hours.

Our aim in this paper is threefold. First, using data from GM (UK) schemes throughout 2016 we sought to understand whether the service is being used and whether this varies by day of week and over time. Second, we sought to understand whether there were differences in the services being used by particular age and gender groups. Third, we aimed to compare the users of extended access appointments to users of general practice core hours. The findings from these exploratory analyses will help inform considerations about extending access to general practice and how such a service might be better targeted to particular population groups.

## METHODS
### Data
The study evaluated the second wave of GM schemes that delivered extended access throughout 2016. Out of GM's 10 Clinical Commissioning Group (CCG) areas, three were already delivering extended access under

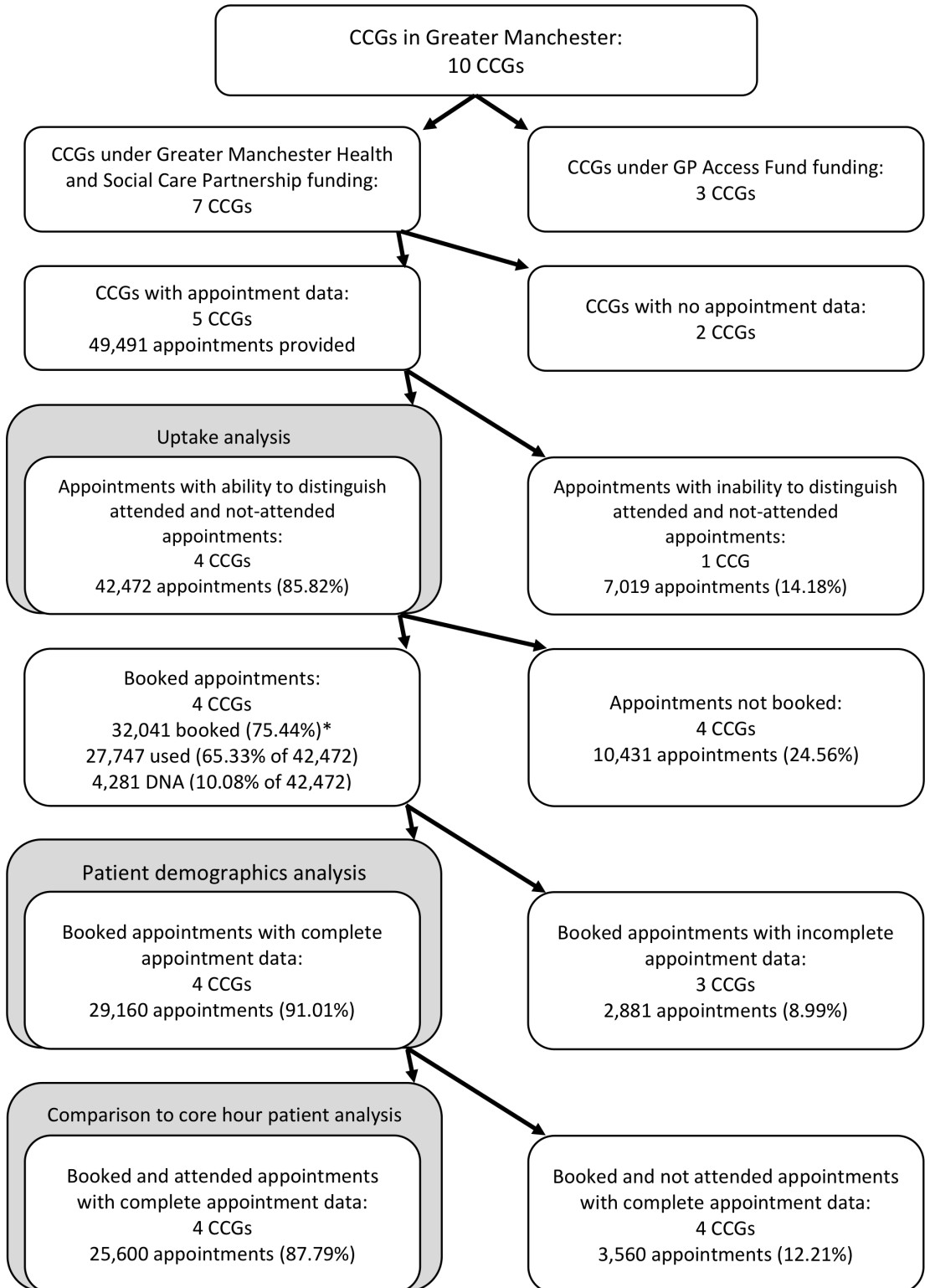

**Figure 1** Flow chart of population coverage. *There are 13 appointments booked that contained no information on attendance. CCG, Clinical Commissioning Group.

the national GPAF scheme during the study period (see figure 1). In line with the GM Health and Social Care Partnership Primary Care Strategy, the other seven CCG areas, not previously in receipt of Wave 1 PMCF or Wave 2 GPAF funding, received financial support from GMHSCP to implement extended access. However, only

five of these seven CCGs were included in the evaluation because two of the CCGs were not captured in the data collection process: one had not implemented the service in 2016; the other did not provide data. A table summarising the five CCG schemes with extended access can be found in online supplementary table S1. The

evaluation was carried out by the National Institute for Health Research Collaboration for Leadership in Applied Health Research and Care Greater Manchester (NIHR CLAHRC GM) and comprised assessments of activity, a process evaluation (seeking to understand the organisational and operational issues that arose in the implementation of the service and how areas addressed these) and assessments of impacts on other sectors of the healthcare system.[13] This study builds on the assessment of activity contained within the evaluation report.

We used administrative data on extended access appointments available, booked and attended in CCG schemes in GM by calendar date over the year 2016 (January to December). The NIHR CLAHRC GM evaluation team requested a minimum data set from each CCG (online supplementary table S2). The aim of this data collection was to inform what new appointment activity was associated with the extended access schemes, whether this new activity was used, and to understand the types of patients using the service (a protocol for the evaluation can be found in online supplementary text S1). Data on each appointment made available were collected and made available by the responsible CCG. CCGs submitted the data to NIHR CLAHRC GM throughout the period. The data included information on date, use (volume available, booked, booked and used), type of appointment (family practitioner or nurse practitioner, prebooked (booked on a previous calendar date) or booked same day) and patient characteristics (the patient's registered practice, age and gender) for booked and attended appointments. Data quality and coverage was dependent on the CCG submitting the data, while NIHR CLAHRC GM could raise instances of incomplete data with CCGs it was up to the CCG to respond. Patient characteristics were limited due to issues of patient confidentiality. To maintain anonymity, patient age was given in 10-year age bands. Deprivation statistics were considered although, due to data confidentiality, we were only able to obtain partial patient postcodes. Since partial postcodes do not enable a suitable deprivation measure to be derived, no deprivation analysis was conducted for the analyses. We were unable to obtain health status or outcomes of patients attending the service.

Appointment data for each CCG area were collected and analysed once a hub became operational. Live dates varied across CCGs, and within CCGs there was variation in hub start dates. A patient residing within their CCG could use any hub services within the CCG.

Data on general practice core hour use were sourced from the GP Patient Survey (GPPS).[14] This is a cross-sectional survey of randomly selected patients from each practice within England. Patients need to be registered with a practice, have an NHS number and be aged 18 and over to be sampled. Patients are asked questions concerning the use of primary care, perceptions of primary care and patient demographics.

## Analysis

All estimation was conducted in Stata (V.14). We assessed uptake, variations in patient demographics for different types of extended access appointments, and made comparisons of extended access appointment users to core hour users as follows:

### Uptake

Patients could book an appointment which could then be attended or not. Focusing on booked appointments alone would give an indication of the demand for the service, while assessments of appointments actually used shed light on actual impacts on service use. We estimated two separate sets of probability models:

1. The probability an appointment was booked.
2. The probability an appointment was booked and attended ('used').

Each model was estimated via probit regressions (see online supplementary text S2 for a technical description of the methods). Regression models were estimated to enable adjustment for multiple predictor variables, for example, CCG effects which may be important given the variation in schemes across CCGs (online supplementary table S1). Probit models were used to estimate differences in the probability (rate) of uptake across categories of the predictor variables. The estimates give absolute percentage point differences across predictor variables which is more informative than relative ORs for understanding the scale of uptake.

To assess whether uptake of the service varied across day of the week we included day of week dummies in the regression, with the estimates providing the differences in the rate of uptake relative to the base week day (Monday). This enabled a test for whether uptake was lower on particular days. To assess whether there was evidence of assimilation to the service we included calendar month dummies, if assimilation is evident the estimates on the calendar month dummies should be monotonically increasing over the period. To account for potential confounding caused by variations in the provision and delivery of services across schemes, we included CCG dummies (fixed effects) in the regression. Robust SEs were estimated to address concerns of heteroscedasticity and clustered at CCG. Estimates are presented as average marginal effects, which give the average percentage point effect on uptake of the variable relative to the base category across the sample.

Analyses were restricted to four CCG schemes due to the inability to differentiate whether a booked appointment had been attended in one CCG (CCG1).

### Analysis of demographics of patients booking an extended access appointment

Our second set of analyses concentrated on booked appointments (both used and not attended) and exploited variations in service delivery to test whether there were variations in patient demographics for particular types of

services. We estimated three separate sets of probability models:

1. The probability an appointment was prebooked versus same day.
2. The probability an appointment was for a GP versus a nurse practitioner.
3. The probability an appointment was used versus not attended (DNA).

In each model the unit of analyses is the appointment, we included dummies for gender (female) and 10-year age bands of the patient booking the appointment, and day of the week of the appointment, month of the appointment and CCG. Each model was estimated via probit regression (see online supplementary text S2 for a technical description of the methods). For all models, estimates are presented as average marginal effects, with SEs clustered at GP practice of the patient booking the appointment. We are unable to identify whether a person booked multiple appointments in the data and therefore assume independence of observations. Schemes that exclusively provided a certain service (eg, only GP appointments) were excluded as a result of uptake of the alternative being necessarily zero.

Missing data on booked appointments were treated as missing at random. To evaluate whether or not missing data might have been associated with the covariates in our models (and hence potentially bias our estimates) we generated a binary missing data indicator for each appointment booked. The binary missing data indicator was coded as 1 where missing data existed for either: age, gender, appointment type, booking type, or the GP practice of the patient booking the appointment. This was regressed via a probit model against day of the week, calendar month and CCG indicators to test whether missing data were related to particular days or months of provision and/or the unit of delivery (CCG).

### Comparisons of extended access appointment users to core hour users

Finally, to inform the potential implications of the service on equity of access to primary care, the demographics of extended access patients who booked and used their appointment were compared with age and gender distributions of patients reporting using their practice within the last year from the January to March 2013 wave of the GPPS.[14] The GPPS is a patient satisfaction survey and as such is not specifically aimed at capturing the volume of use of general practice services, however, in the absence of data on general practice use and demographics of users in England, the survey presents the most appropriate data set for comparison. Patients reporting in the January to March 2013 wave were selected to proxy for patient demographics in usual care. Later waves of the GPPS could cause potential bias as patients reporting having used primary care may have used an extended access service which had been available in areas of GM since 2014. The GPPS sample was reduced to patients registered with a practice within one of the four CCG areas under evaluation and further reduced to those patients using primary care over the last year. These patients were determined via responses to question 1 of the survey: '*When did you last see or speak to a GP from your GP surgery?*', with patients reporting '*In the past 3 months*', '*Between 3 and 6 months ago*' and '*Between 6 and 12 months ago*'. The age groupings of these respondents did not align with the age groupings in the appointment data. As such, to ease comparisons, population pyramids are presented to give an overall feel for similarities and/or differences between the two groups of patients. The GPPS may suffer from response bias. We therefore weight practice patient use measures in this survey by the sampling weights provided with the GPPS data set for non-response related to age and gender.[14]

### Patient and public involvement

A panel of patients reviewed an earlier draft of the protocol, where feasible the protocol was adapted in light of patient feedback. The response from patients in relation to the activity analyses focused on being able to capture variations by day of week and type of appointment, no changes in data capture were necessary following patient responses as these concerns were anticipated to be addressed in the appointment activity analyses (see online supplementary text S3).

## RESULTS
### Uptake

A total of 51 806 extended access appointments were provided throughout 2016 by the five CCGs under the GM scheme (table 1), 41.07 appointments per 1000 of the population (the population covered by these CCGs is 1 261 326). Of these, across two CCGs, 2315 were 'blocked' and not bookable by patients resulting in a total of 49 491 bookable appointments. CCG1 was removed from the analysis since no data on whether the patient attended were provided leaving a sample of 42 472 (49 491 appointments – 7019 CCG1 appointments). 75.44% of 42 472 bookable appointments were booked and 65.33% were booked and attended. Provision and uptake varied across the CCGs, with CCG2 dominating provision and scale per 1000 of the population and CCG5 having the highest percentage use. The types of appointments differed across the CCG schemes (table 2). Two of the four schemes provided both GP and nurse appointments, the other GP only, and one scheme provided only prebookable appointments. Of all appointments booked, 53.27% were prebooked and 81.42% were for a GP appointment.

Figure 2 plots appointments used, appointments booked and not attended ('DNA'), and appointments not booked over the 2016 calendar year. Use appears to have improved slightly over time. A total of 15 575 appointments (36.67%) occurred on a Saturday, and Sunday was the second most active day (5878, 13.84%) (figure 3). 52.93% of appointments on a Sunday were booked (46.73% used), and 75.06% (63.94% used) on a Saturday. The percentage

**Table 1** Volume of appointments bookable, booked, used and did not attend (DNA)

| Provider | Provided (rates per 1000, population (P) size) | Blocked | Available | Not booked* | Booked* (% of available) | DNAs† (% of available) | Used* (% of available) |
|---|---|---|---|---|---|---|---|
| CCG1 | 7019 (23.23 per 1000, P=302 163) | 0 | 7019 | 1500 | 5519 (78.63) | | |
| CCG2 | 32 693 (143.77 per 1000, P=227 391) | 1911 | 30 782 | 7437 | 23 345 (75.84) | 2959 (9.61) | 20 380 (66.21) |
| CCG3 | 3637 (14.63 per 1000, P=248 667) | 0 | 3637 | 1587 | 2050 (56.37) | 202 (5.55) | 1848 (50.81) |
| CCG4 | 5600 (22.92 per 1000, P=244 325) | 0 | 5600 | 1137 | 4463 (79.70) | 758 (13.54) | 3699 (66.05) |
| CCG5 | 2857 (11.96 per 1000, P=238 780) | 404 | 2453 | 270 | 2183 (88.99) | 362 (14.75) | 1820 (74.19) |
| All schemes | 51 806 (41.07 per 1000, P=1 261 326) | 2315 | 49 491 | 11 931 | 37 560 (75.89) | | |
| Schemes CCG2–CCG5 | 44 787 (46.84 per 1000, P=956 163) | 2315 | 42 472 | 10 431 | 32 041 (75.44) | 4281 (10.08) | 27 747 (65.33)‡ |

CCG5 blocked are for administrative purposes.
*Booked are appointments that were used (can be either prebooked or booked same day).
†CCG1 did not record DNAs, we assume all attended for the Greater Manchester (GM) % booked and used; missing data on DNA evident for six appointments in CCG2, six in CCG4 and one in CCG5.
‡As a percentage of appointments where attendance was captured (42 472 appointments: 49 491 appointments minus 7019 CCG1 appointments).
CCG, Clinical Commissioning Group.

of appointments booked on weekdays was generally higher at 73.29%–86.06% (65.94%–75.46% booked and used) compared with weekend appointments at 52.94% (Sundays) and 75.06% (Saturdays) (46.73% (Sundays) and 63.94% (Saturdays) booked and used).

The results from the probability models of appointments booked and appointments booked and used are contained in table 3 (unadjusted analyses are presented in online supplementary table S3). Estimates for the probability models are interpreted as the percentage point difference

in the rate of the outcome measure relative to the base category. For example, the estimate for Tuesday in the 'Probability model for appointment booked' model is interpreted as the rate of appointments booked on a Tuesday was 11.45 percentage points higher than the rate of appointments booked on a Monday (the base category). The results broadly confirm the summary statistics with Monday having the lowest percentage of appointments booked of other weekdays (Tuesday to Friday), and Sunday having the lowest percentage of appointments booked overall (−18.93

**Table 2** Volume of appointments by appointment type (GP/nurse) and booking type (prebooked/same day)

| Provider | Appointments available | GP* | % GP | Nurse* | % Nurse | Appointments booked | Prebooked† | % Prebooked | Same day† | % Same day |
|---|---|---|---|---|---|---|---|---|---|---|
| CCG2 | 30 782 | 24 252 | 78.79 | 6530 | 21.21 | 23 345 | 9805 | 42.00 | 13 540 | 58.00 |
| CCG3 | 3637 | 3637 | 100.00 | 0 | 0.00 | 2050 | 524 | 25.56 | 1526 | 74.44 |
| CCG4 | 5600 | 5255 | 93.84 | 0 | 0.00 | 4463 | 2460 | 55.12 | 2003 | 44.88 |
| CCG5 | 2453 | 1438 | 58.62 | 1015 | 41.38 | 2183 | 2180 | 99.86 | 0 | 0.00 |
| Total | 42 472 | 34 582 | 81.42 | 7545 | 17.76 | 32 041 | 14 969 | 46.72 | 17 069 | 53.27 |

Prebooked appointments are appointments booked on a previous calendar date. Information on prebooked or same day only available for booked appointments.
*GP/nurse contains missing for CCG4 (345 appointments).
†CCG5 has three appointments with no data on booked time.
CCG, Clinical Commissioning Group; GP, general practitioner.

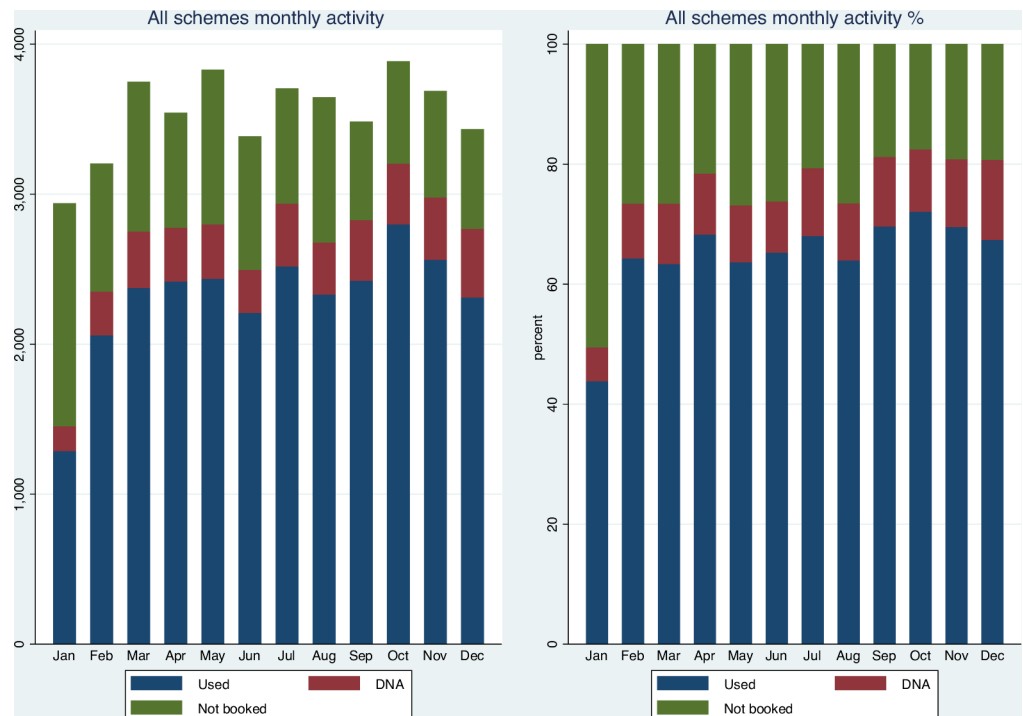

**Figure 2** Volume and percentage of appointments used, DNA and not booked by month, all schemes. Excludes CCG1 activity. DNA: appointments booked but not attended. Used: appointments booked and attended. CCG, Clinical Commissioning Group.

percentage points compared with Monday (95% CI −35.03 to −2.83)). The results are similar for appointments used. The percentage of appointments booked and used over the year was greater for all months compared with January and appears to be greater in the second half of the year,

however, the increase over time is small and inconsistent with lower rates of appointments booked and used in May, June, August, November and December compared with neighbouring months. There were differences in uptake at CCG level with all CCGs having lower percentages of

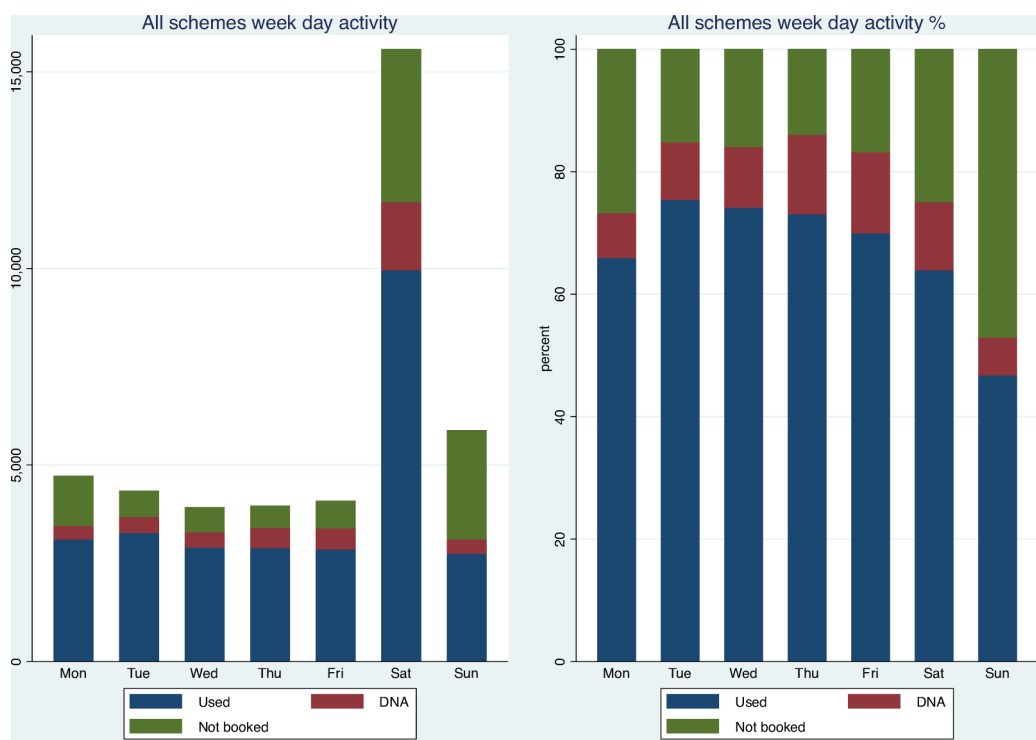

**Figure 3** Volume and percentage of appointments used, DNA and not booked by day of week, all schemes. Excludes CCG1 activity. DNA: appointments booked but not attended. Used: appointments booked and attended. CCG, Clinical Commissioning Group.

**Table 3**  Rates (%) and probability models for appointments booked and appointments booked and used

| | % Booked | Probability model for appointment booked (95% CI) | % Booked and used | Probability model for appointment booked and used (95% CI) |
|---|---|---|---|---|
| **Day of week** | | | | |
| Monday (base category) | 73.29 | | 65.94 | |
| Tuesday | 84.84 | 11.45 (8.51 to 14.38) | 75.46 | 9.60 (7.45 to 11.75) |
| Wednesday | 84.06 | 11.06 (5.89 to 16.24) | 74.14 | 8.45 (3.32 to 13.58) |
| Thursday | 86.06 | 12.58 (10.05 to 15.10) | 73.07 | 7.04 (4.64 to 9.44) |
| Friday | 83.21 | 10.05 (7.12 to 12.98) | 69.96 | 4.19 (0.88 to 7.51) |
| Saturday | 75.06 | −0.10 (−8.44 to 8.24) | 63.94 | −3.47 (−10.97 to 4.01) |
| Sunday | 52.94 | −18.93 (−35.03 to −2.83) | 46.73 | −18.07 (−32.46 to −3.68) |
| **Calendar month** | | | | |
| January (base category) | 49.52 | | 43.87 | |
| February | 73.45 | 22.73 (13.25 to 32.20) | 64.33 | 19.80 (13.15 to 26.45) |
| March | 73.45 | 20.92 (11.80 to 30.04) | 63.39 | 17.56 (12.25 to 22.88) |
| April | 78.48 | 26.84 (11.48 to 42.20) | 68.31 | 23.42 (11.91 to 34.94) |
| May | 73.17 | 22.76 (7.90 to 37.62) | 63.69 | 19.21 (8.32 to 30.10) |
| June | 73.82 | 22.45 (5.40 to 39.50) | 65.31 | 20.45 (7.37 to 33.52) |
| July | 79.37 | 29.08 (16.09 to 42.07) | 68.05 | 24.13 (13.98 to 34.27) |
| August | 73.52 | 22.60 (4.12 to 41.07) | 64.00 | 19.11 (4.95 to 33.28) |
| September | 81.25 | 30.54 (12.66 to 48.43) | 69.64 | 25.26 (11.19 to 39.32) |
| October | 82.25 | 32.26 (15.89 to 48.62) | 72.12 | 28.11 (15.33 to 40.89) |
| November | 80.85 | 29.71 (12.79 to 46.62) | 69.56 | 24.75 (11.59 to 37.91) |
| December | 80.74 | 29.42 (6.82 to 52.01) | 67.40 | 23.01 (5.75 to 40.26) |
| **CCG scheme** | | | | |
| CCG1* | | | | |
| CCG2 | 75.48 | −16.63 (−20.09 to −13.17) | 66.21 | −10.43 (−13.17 to −7.68) |
| CCG3 | 56.37 | −32.02 (−35.24 to −28.80) | 50.81 | −23.85 (−26.36 to −21.35) |
| CCG4 | 79.70 | −11.28 (−14.69 to −7.87) | 66.05 | −9.97 (−12.55 to −7.38) |
| CCG5 (base category) | 88.99 | | 74.19 | |
| Sample size | 42 472 | 42 472 | 42 472 | 42 472 |

Appointments booked are appointments booked, appointments booked and used are appointments that were booked and subsequently attended. Probability models are probit regressions of appointment status against day of week, calendar month and CCG scheme. SEs are clustered at the CCG level. Estimates are presented as average marginal effects which give the percentage point effect of the variable relative to the base category.
*CCG1 did not provide data to enable identification of whether a booked appointment was subsequently attended so does not feature in the analysis.
CCG, Clinical Commissioning Group.

appointments booked and used compared with CCG5. For comparison, online supplementary table S4 replicates the analyses with estimation via logistic regression and confirms the same sets of estimates that are significantly different from zero in the probit regressions are significantly different from one in the logistic regression. The relative size of magnitude of the estimates is also similar across models.

To test whether the percentage of appointments booked and used on a Sunday improved over and above the general trend over the period we replicated our analysis with the inclusion of Sunday calendar month interaction terms (online supplementary table S5). The interaction terms were not significantly different from zero, suggesting any change over time was similar to that observed from Monday to Saturday in the data.

### Analysis of demographics of patients booking an extended access appointment

The analyses concerning variations in patient demographics for particular types of appointments are limited to booked appointments (used and DNA) (figure 1). Of the 37 560 booked appointments, 5519 were removed due to data reporting by CCG1, giving 32 041 appointments over the remaining four CCGs. Missing data on appointment type, booking type, and patient's gender and age and GP practice amounted to a further 2881 appointments being excluded from the analyses (online supplementary table S6), giving a sample of 29 160 booked appointments. Missing data varied by CCG but were similar within CCGs by hub. CCG1 provided 100% complete data. Missing data appear to be associated with calendar month, day of week

(Wednesday and Sunday) and CCG (online supplementary table S7).

Of the 29 160 booked appointments with complete data, 46.43% were prebooked, 82.54% were for a GP and 12.17% resulted in a patient not attending (online supplementary table S8). 58.60% of booked appointments were made by female patients. 55.91% of users were aged under 39 (71.48% under 49). Estimates from our second set of analyses concerning patient demographics for types of appointments booked are contained in table 4. Across all models there were significant CCG effects; this is likely to reflect variations in service delivery due to the variations in relative scale of booking type and appointment type (table 2).

58.16% of prebooked appointments were booked by female patients and 58.09% of same-day appointments were booked by female patients (online supplementary table S9), we found no significant difference in the type of appointment booked by females compared with males (−0.73 percentage points less likely to be prebooked compared with males, 95% CI −2.01 to 0.55). Appointments booked for patients in age groups 10–19 to 80–89 were more likely to be prebooked in comparison to age group 0–9 years. Tuesday to Sunday appointments were more likely to be prebooked compared with Monday. GP appointments were less likely to be prebooked in comparison to nurse appointments (−12.97 percentage points, 95% CI −19.49 to −6.45). Patients were more likely to turn up to a same-day booked appointment than appointments that had been booked on an earlier calendar date (DNA rates were higher for prebooked appointments by 19.08 percentage points, 95% CI 16.11 to 22.04).

57.56% of GP appointments were booked by females (64.59% of nurse appointments), appointments booked by females were significantly less likely to be with a GP compared with males (−4.29 percentage, 95% CI −6.42 to −2.16). Appointments booked were less likely to be with a GP as the age of the patient booked for the appointment increases compared with age group 0–9. There was limited evidence that appointments booked on specific weekdays were more or less likely to be for GP appointments. Appointments that were prebooked were less likely to be GP appointments. Appointments that were not attended were no more or less likely to be GP appointments (1.97 percentage points higher rate for GP appointments compared with used appointments (95% CI −0.39 to 4.32)).

58.51% of used appointments were booked by females and 59.13% of not attended (DNA) appointments were booked by females, no significant difference was found in the likelihood that appointments booked by females were not attended compared with males (−0.32 percentage points lower likelihood of not attending (95% CI −1.40 to 0.77)). Appointments booked by age group 20–29 had a higher likelihood of not being attended compared with age group 0–9 though this effect is small (0.32 percentage points (95% CI 1.65 to 4.74)). Appointments booked by age groups above the age of 40 were significantly more likely to be used compared with age group 0–9. Appointments booked for a Thursday, Friday or Saturday were more

likely to be not attended compared with those booked on a Monday.

## Comparisons of extended access appointmentusers to core hour users

To compare patient demographics of extended access users to core hour users we focus on appointments that were used. In total, 84.47% of appointment users were aged under 60 years (table 5 and figure 4). Patients booking and attending extended access appointments were compared with patients reporting attendance at their practice within the past year in the GPPS and the population registered with a practice. To better reflect the sampling frame of the GPPS (patients aged 18 and over), figure 5 and online supplementary table S10 provide the percentages of users in extended access appointments from age 20+. 80.40% of extended access appointment users aged 20+ were under 60 years, while the equivalent percentage in the GPPS data lies above age 65. Users of extended access appointments tended to be younger in comparison to core hour patients with a more equal distribution of users aged between 20 and 59 compared with GPPS core hour users. For female users, the peak in the distribution of extended access users occurred at ages 20–29, compared with 45–54 for users in the GPPS. As the age bands do not align in the data sets, these differences could not be tested, and comparisons should be treated with caution.

## DISCUSSION
### Principal findings
Our primary analyses concerned the proportion of extended access appointments used. We found evidence of spare capacity in the extended access service and limited evidence that use improved over the calendar year. The proportion of appointments used was lowest on Sundays. Secondary analyses investigated types of appointments booked (both booked and used and booked and not used) and the patient demographics of users of the service. There were variations in types of patients booking particular types of extended access appointments. Prebooked appointments were relatively more likely among age group 10–89 compared with age group 0–9. GP appointments were increasingly less likely to be booked with age. Appointments booked on Thursday to Saturday were relatively more likely to result in non-attendance compared with Mondays. Prebooked appointments were less likely to be with a GP and more likely to result in non-attendance compared with same-day appointments. In comparison to core hour patients, we found some evidence that patients using extended access appointments were of a younger age relative to patients using primary care services before the scheme. The most obvious difference was found in females aged 20–29 who accounted for almost a quarter of all female extended access appointment users (aged 20+).

**Table 4** Patient demographics by booking and appointment type

| | Prebooked versus same day* (95% CI) | GP versus nurse† (95% CI) | DNA versus used (95% CI)‡ |
|---|---|---|---|
| **Gender** | | | |
| Male (base category) | | | |
| Female | −0.73 (−2.01 to 0.55) | −4.29 (−6.42 to −2.16) | −0.32 (−1.40 to 0.77) |
| **Age band (years)** | | | |
| 0–9 (base category) | | | |
| 10–19 | 13.39 (10.14 to 16.64) | −19.37 (−25.83 to −12.91) | 0.66 (−1.08 to 2.40) |
| 20–29 | 14.45 (11.28 to 17.63) | −26.45 (−32.34 to −20.56) | 0.32 (1.65 to 4.74) |
| 30–39 | 14.53 (11.79 to 17.26) | −29.45 (−35.79 to −23.10) | 0.97 (−0.79 to 2.74) |
| 40–49 | 15.03 (11.72 to 18.35) | −31.86 (−39.36 to −24.36) | −3.20 (−4.95 to −1.44) |
| 50–59 | 17.71 (14.52 to 20.90) | −31.89 (−39.46 to −24.32) | −7.04 (−9.07 to −5.01) |
| 60–69 | 13.25 (9.09 to 17.41) | −34.25 (−43.65 to −24.86) | −12.05 (−14.29 to −9.81) |
| 70–79 | 12.60 (8.78 to 16.42) | −37.40 (−47.70 to −27.10) | −10.20 (−13.16 to −7.23) |
| 80–89 | 14.00 (9.45 to 18.55) | −41.26 (−52.96 to −29.55) | −7.86 (−10.91 to −4.81) |
| 90+ | 8.83 (−5.79 to 23.45) | −41.39 (−63.28 to −19.49) | – |
| **Day of week** | | | |
| Monday (base category) | | | |
| Tuesday | 20.49 (14.45 to 26.53) | −0.63 (−5.05 to 3.79) | −0.57 (−2.12 to 0.98) |
| Wednesday | 27.02 (23.30 to 30.74) | 3.32 (0.01 to 6.63) | −0.59 (−2.32 to 1.14) |
| Thursday | 24.39 (19.00 to 29.78) | −2.24 (−7.15 to 2.67) | 2.77 (1.33 to 4.21) |
| Friday | 25.59 (20.88 to 30.30) | 1.99 (−1.61 to 5.59) | 3.47 (1.62 to 5.32) |
| Saturday | 24.23 (18.91 to 29.56) | 2.32 (−2.61 to 7.24) | 2.22 (0.66 to 3.77) |
| Sunday | 24.98 (20.62 to 29.33) | 0.75 (−2.31 to 3.81) | −0.37 (−2.10 to 1.37) |
| **Appointment type** | | | |
| Nurse (base category) | | | |
| GP | −12.97 (−19.49 to −6.45) | – | 1.11 (−0.63 to 2.85) |
| **DNA status** | | | |
| Used (base category) | | | |
| DNA | 19.08 (16.11 to 22.04) | 1.97 (−0.39 to 4.32) | – |
| **Booking type** | | | |
| Same day (base category) | | | |
| Prebooked | – | −7.46 (−12.29 to −2.62) | 8.48 (7.24 to 9.73) |
| **CCG scheme** | | | |
| CCG2 | −14.90 (−23.30 to −6.50) | 15.22 (7.37 to 23.07) | 2.94 (0.61 to 5.27) |
| CCG3 | −27.19 (−36.88 to −17.51) | – | 0.96 (−2.26 to 4.18) |
| CCG4 | Base | – | 0.15 (−2.76 to 3.05) |
| CCG5 | – | Base | Base |
| Sample size | 27 035 | 23 429 | 29 120 |

Prebooked appointments are appointments booked on a previous calendar date. Probability models are probit regressions of appointment against day of week, calendar month, appointment characteristics, patient age and gender, and CCG scheme. SEs are clustered at the patient's GP practice level. Month estimates omitted from the output but available on request. Estimates are presented as average marginal effects which give the percentage point effect of the variable relative to the base category.

*CCG5 had 100% prebooked and GP appointments (n=2125 with complete data) and was omitted from this analysis giving a sample of 29 160–2125=27 035.

†CCG4 had 100% GP appointments (n=3704 with complete data) and CCG3 had 100% GP appointments (n=2027 with complete data) and both were omitted from this analysis giving a sample of 29 160–3704–2027=23 429.

‡All appointments booked by patients age 90+ were attended (40 appointments), these patients are omitted from the analysis giving a sample of 29 160-40=29 120.

CCG, Clinical Commissioning Group; GP, general practitioner.

**Table 5** Demographic distributions of extended access users

| | Males | | | Females | | | All |
| Age | Frequency | % | Cumulative % | Frequency | % | Cumulative % | Cumulative % |
| --- | --- | --- | --- | --- | --- | --- | --- |
| 0–9 | 1436 | 13.53 | 13.16 | 1387 | 9.26 | 9.26 | 11.03 |
| 10–19 | 1018 | 9.59 | 23.12 | 1489 | 9.94 | 19.20 | 20.82 |
| 20–29 | 1446 | 13.62 | 36.74 | 2897 | 19.33 | 38.53 | 37.78 |
| 30–39 | 1575 | 14.84 | 51.58 | 2513 | 16.77 | 55.30 | 53.75 |
| 40–49 | 1617 | 15.23 | 66.81 | 2435 | 16.25 | 71.55 | 69.58 |
| 50–59 | 1690 | 15.92 | 82.73 | 2123 | 14.17 | 85.72 | 84.47 |
| 60–69 | 1144 | 10.78 | 93.51 | 1273 | 8.50 | 94.22 | 93.91 |
| 70–79 | 481 | 4.53 | 98.04 | 624 | 4.16 | 98.38 | 98.23 |
| 80–89 | 197 | 1.86 | 99.90 | 215 | 1.43 | 99.81 | 99.84 |
| 90+ | 12 | 0.11 | 100.00 | 28 | 0.19 | 100.00 | 100.00 |
| Total | 10 616 | 100.00 | | 14 984 | 100.00 | | |

## Strengths and weaknesses

The study has a number of strengths and weaknesses. Strengths lie in the unique data set which enabled a detailed look at uptake and users of the service and how this varied with types of appointments provided. In addition, our results are focused on several GM schemes offering variation in delivery which enables comparisons of different service models. However, weaknesses are that the findings may not be generalisable to other areas that may provide an extended access service under alternative specifications and/or populations than those detailed in this study. Missing data on patient age, gender and registered practice resulted in a loss of 8.99% of appointments and this was found to vary with day of week, calendar month and CCG. Although data issues were flagged by the research team, we could not enforce areas to provide complete data. We were unable to obtain patient deprivation/ethnicity characteristics to enrich understanding of service use. We were also unable to identify whether there are repeat patients in the data, this may bias the comparison of users of the extended access service to those in core hours if repeated use is associated with gender and/or age. We were not able to identify whether an appointment was cancelled and as such, we assume this was subsequently made available. The GPPS comparison is particularly flawed because the measure of core hour use is self-reported and thus may be affected by recall error, children are not included in the survey and age bands did not match with those obtained for appointment data making it not possible to draw direct comparisons of age groups. Finally, access depends on a range of factors, and without further details on CCG provision related to

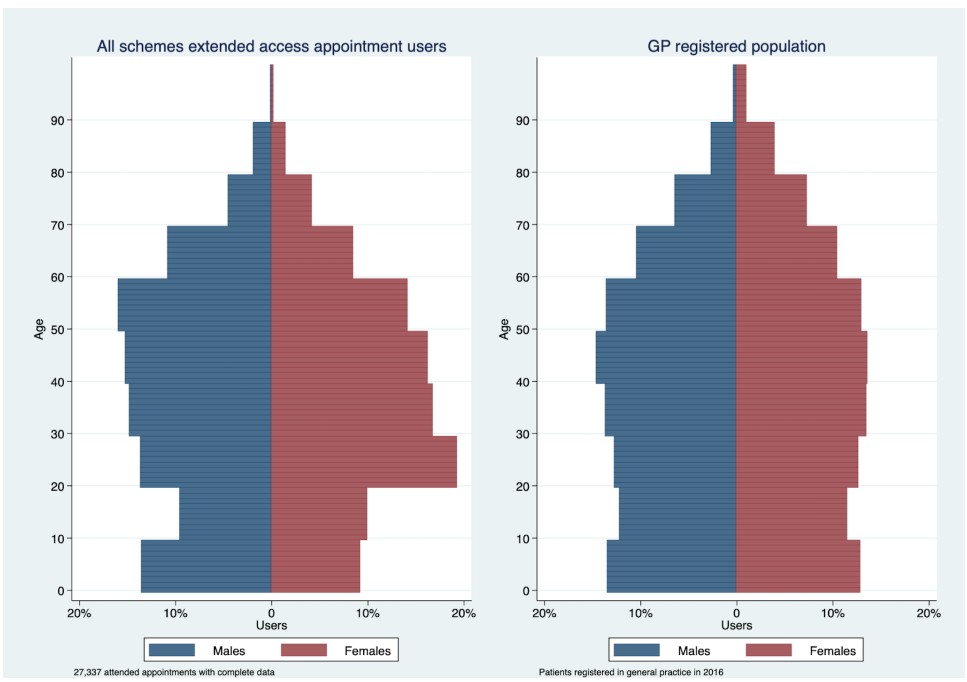

**Figure 4** Profile of extended appointment users by age and gender. GP, general practitioner.

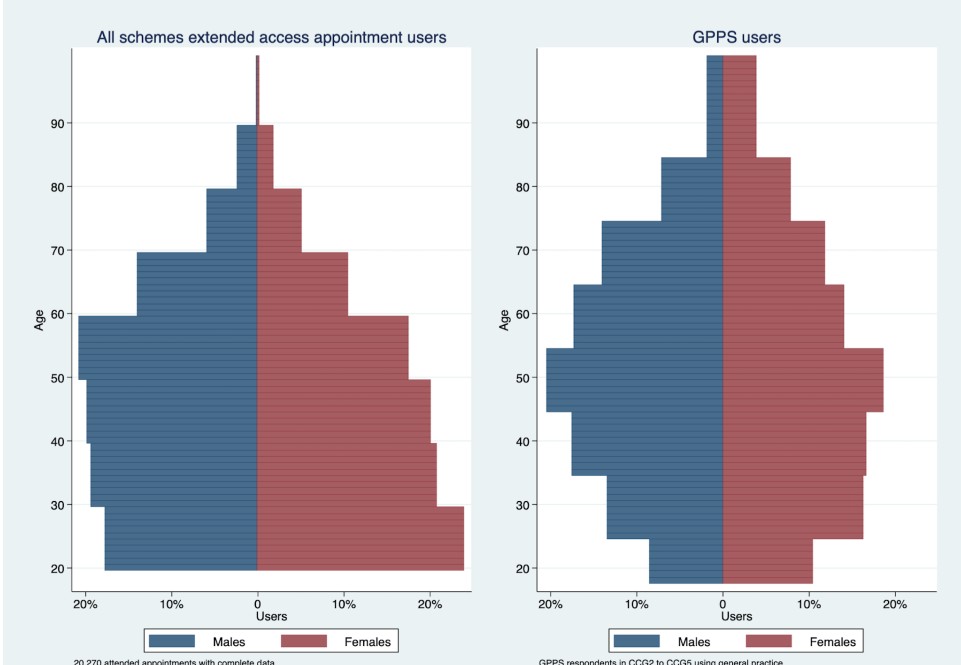

**Figure 5** Profile of extended appointment users and core hour users, by age and gender. GPPS data are patients reporting having spoken to or seen a general practitioner (GP) from their practice in the past 12 months. CCG, Clinical Commissioning Group; GPPS, GP Patient Survey (January to March 2013 wave).

awareness campaigns and buy-in from providers, we can only distinguish such effects by the CCG fixed effects adjustment factor in our models.

### Implications

The presence of spare capacity for a service seeking to improve access to meet excess demand in core hours raises the question of whether or not extended access appointments really are improving access to primary care. However, of the appointments that were used, our findings suggest that the service may be either (1) meeting previously unmet need for younger age groups, (2) meeting previously inappropriately met need should these patients have proceeded to use alternative health services such as emergency departments at hospital, or (3) diverting demand for core hours for this age group, thereby potentially reducing pressure in core hours. It is likely that a mixture of these factors is playing a role. We find 46.43% of appointments used were prebooked which may signify a large proportion of patients are using the service for non-urgent health needs which may suggest (1) or (3) plays an important role. 53.57% of appointments booked and used were same-day appointments which may suggest (2) may also be an important factor. More research is needed to understand the relative scale of these factors, such information would give important insights into the value and efficiency of the service. These factors are on the assumption that patients accessing the service had a need for seeking healthcare services, should patients referred to the service not have a need for healthcare then service use may reflect supply-induced demand.

The reasoning behind spare capacity in the scheme may be informed by three domains of access: availability, acceptability, affordability.[15] These domains of access highlight the complex contributory factors at play in access, involving both patient and practitioner. The extended access schemes provide additional *availability* in the sense of new appointments. However, new appointments alone may not be sufficient to improve access, should patients or practitioners not be prepared to *accept* the service (this could be due to cultural factors and/or patient preferences for example) or should patients not be aware of the service. A process evaluation of the service is contained in the evaluation report of the service and highlights the variation in approaches taken to communicate the service to patients.[13] Identification is needed of whether poor rates of use are due to a lack of patient demand or, of particular concern in areas where practice referrals are made, whether a lack of receptionist/practitioner buy-in is influencing patient (and receptionist) awareness of the service (see eg, Healthwatch Manchester, 2017[16]). Policy needs to consider these additional aspects of access to facilitate use and adapt provision accordingly.

Cost-effectiveness analysis seeks to understand whether a service is an efficient use of healthcare resources, this is performed by identifying the opportunity cost of delivering the service (the net benefits of services that could have been provided by allocating the resources elsewhere). While we do not compare the service to a counterfactual use of resources in this study (such as expanding core hours), our findings shed light on a number of aspects relevant for determining cost-effectiveness of the extended access service. First, it is likely that extended access services will result in some positive measure of benefit to patients. Second, a large proportion

of extended appointments are prebooked, suggesting a large amount of activity in the service is for non-urgent care. As such, existing evaluations, which have focused on emergency care implications, paint only a partial picture of the impacts of extended access. Third, only 65.33% of appointments available were booked and used which raises concerns relating to how efficient the service is. Providers seeking to determine the types of services available under extended access schemes may consider several factors to more efficiently provide the service in relation to our results: (1) While the proportion of appointments used was lowest on a Sunday, the volume of appointments used was similar to those during the week. This suggests the proportion of appointments used on a Sunday can be improved by reducing the volume of Sunday appointments. (2) A completely prebookable service may reduce access for the very young and result in higher rates of appointments booked and not attended. (3) Nurse appointments may help address issues of staffing. DNAs were no greater for this type of appointment. However, GP appointments were relatively more concentrated among age group 0–9 and males suggesting a potential need for a GP service to address the needs of younger age and male patients.

The evaluation of the national pilots has found uptake of 71% with reportedly lower uptake on weekends, particularly on Sundays.[8] A more detailed evaluation of three schemes under the PMCF pilots found uptake varied by type of service (same day or prebookable) available.[11] Weekend uptake was higher in a scheme providing prbookable and same-day appointments (87% on Saturdays and 78% on Sundays) compared with two schemes providing only urgent/same-day appointments (25% and 18% and 48% and 41%). The demographics of the scheme providing both prebookable and same-day appointments comprised 3% aged under 5, and 57% aged between 20 and 65. The findings in this paper found schemes with prebookable and same-day appointments had a lower rate of booked appointments than schemes with only same-day appointments (the proportion of appointments booked in CCG2–CCG4 are lower than that of CCG5 which provided only same-day appointments (table 2)). We also found uptake was lower on Sundays (also in line with the national evaluation). However, in contrast, we found age group 20–29 to be the largest group of patients using the service (17%) and a higher proportion of patients from the age group 20–69 (73% compared with 57% aged 20–65). These differences may be due to differences in the patient population, breadth of week covered or scale of the schemes (the relevant scheme evaluated by Windrum et al[11] was much smaller with three practices and one hub). Evaluations of extended access appointments for urgent care in Sheffield, UK, found users were predominantly female (60.0%) and aged under 60 (85.5%), 19.0% were aged under 5 and 29.3% were aged under 16; users were also concentrated in more deprived areas.[12] These findings

may reflect the urgent care service delivered in comparison to the current study which offered both same-day and prebookable appointments.

## Future work

We have not been able to identify implications for core hours or assess whether deprivation or ethnicity is a leading factor of appointment use. Further research is needed to investigate the impacts on demand for core hour services and for deprived populations to shed light on potential implications for access.

An assessment of the types of treatments and conditions being presented at extended access appointments may help inform impacts of the service on patient health and the scale of potential supply-induced demand.

Given the pilot nature of these schemes, uptake may have also reflected supply-side buy-in, and reservations concerning promotion of the schemes may have arisen due to uncertainty in future funding. Future evaluations of uptake, once clarity is provided on long-term securement of the extended access service, may help identify the scale of this possibility.

**Contributors** WW is the guarantor for the work. WW cleaned the data, conceived and conducted the analysis, interpreted the data and composed the first draft of the work and proceeding revisions. LA critically revised the manuscript, participated in the design of the study and approved the final version to be published. PN and KR contributed to interpretation of data, revised the initial and subsequent drafts of the paper critically for content and approved the final version to be published. CO cleaned the activity data in preparation for the analyses, assisted in interpretation of the findings and approved the final version to be published. NR contributed to interpretation of data, revised earlier drafts of the paper and approved the final version to be published. DH designed and led the overall project, was directly involved in interpretation of data and approved the final version to be published. All authors had full access to all of the data (including statistical reports and tables) in the study and can take responsibility for the integrity of the data and the accuracy of the data analysis.

**Funding** This project was funded by the National Institute for Health Research Collaboration for Leadership in Applied Health Research and Care (NIHR CLAHRC) Greater Manchester and the Greater Manchester Health and Social Care Partnership. The NIHR CLAHRC Greater Manchester is a partnership between providers and commissioners from the NHS, industry and the third sector, as well as clinical and research staff from the University of Manchester.

**Disclaimer** The views expressed in this article are those of the author(s) and not necessarily those of the NHS, NIHR or the Department of Health and Social Care.

**Competing interests** WW reports grant funding for other work from the Department of Health Policy Research Programme and NIHR.

**Patient consent for publication** Not required.

**Ethics approval** The study was reviewed by the University of Manchester's Internal University Research Ethics Committee and approved under the low-risk procedure and thus did not require UREC approval (ID: AMBS/16/05). The study involves anonymised administrative data and did not impact on the type of care patients received.

**Provenance and peer review** Not commissioned; externally peer reviewed.

**Data availability statement** No additional data are available.

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
