## [Reviewer comments · BMJ Open]

ARTICLE DETAILS

TITLE (PROVISIONAL)	An investigation of the demand for a 7-day (extended access) primary care service: an observational study from pilot schemes in England
AUTHORS	Whittaker, William; Anselmi, Laura; Nelson, Pauline; O'Donnell, Caroline; Ross, Natalie; Rothwell, Katy; Hodgson, Damian

VERSION 1 – REVIEW

REVIEWER	Professor Shona J. Kelly Sheffield Hallam University United Kingdom
REVIEW RETURNED	02-Jan-2019

GENERAL COMMENTS	Re: bmjopen-2018-028138 Thank you for the opportunity to review this paper that investigated the use of extended access to primary care services in Greater Manchester. The findings are extremely interesting but some work is needed to make the methodology accessible for a non-statistical audience. 1. The biggest issue the use of a probit model. Why probit? This type of regression is very uncommon in public health/epidemiology. My statistics is at an moderately advanced level and my understanding from the explanation offered by my statistician is that it is used when there are a lot of empty cells. I presume that you had the same problem as we did on a similar set of data and had to deal with a lot of missing data and some appointment slots (on Sundays) that were not taken up? The explanation in the supplementary file is not understandable except by a statistician. You need to find a way to explain this type of regression. Most of the readers of this paper will be familiar with logistic regression so a comparison with it would be a good place to start your explanation of a probit model. You also need to explain how to interpret the tables. For example in Table 3, the "appointment booked" column - how does one interpret the values - the title says "rates", is the usual number of appointments per xxxx patients?2. The quote in the introduction should have the reference at the end removed as it is confusing3. Define acronyms the first time they are used - e.g. 'GM'4. I would have put summary table 1 in the text and put your table 1 in an appendix. Knowing which are nurse or doctor appointments is important for interpreting the findings.5. you need to explain in methods that the data on core hour practice use is patient satisfaction data and it is pretty much useless
--

	6. Were these practice nurse appointments or advanced nurse practitioner appointments? 7. I'm not sure all the detail on the Hubs and CCGs is relevant to the general audience reading this paper. I'd remove all of that as people can read your full online report which you can reference. 8. A comment in the discussion about why the CCGs could get away without providing data would be illuminating. 9. The use of white text on blue in figure 1 is too hard to read. 10. Some of the figures need more footnotes or information in the titles. e.g. Figure 2 needs n(%) 11. Figure 4 needs a note about why the under 18's are missing in the right hand chart. They would also be more comparable if the two charts were scaled the same. I think the left-hand chart is key for the discussion. 12. It is not usual to include the contract as an appendix. You could refer the reader to an online version. It is also confusing because this paper doesn't address all of the outcomes listed in the contract. 13. Some general statements about data accessibility and quality would be useful. Why isn't data easily available? Why can't the CCGs provide the postcode derived deprivation score? Why don't we have better demographics?
--	---

REVIEWER	Louis Levene University of Leicester, United Kingdom
REVIEW RETURNED	13-Jan-2019

GENERAL COMMENTS	General points: Overall, this paper covers an important and interesting topic. The authors have set out a good research question and used sound methodology to answer it (although I could not identify any issues, it may help to have a statistician double-check it as I am not a statistical expert). They have interpreted their results judiciously and have considered the key points and implications. Specific points (apologies for seeming pedantic):  1. Dates covered in 2016 (page 6, line 57). Assuming it is 1 Jan to 31 Dec- just need to make this clear. 2. Outcomes- primary vs. secondary. This is made clear in the abstract and introduction, but not at the start of the discussion (principal findings). 3. Core hours (page 4, line 17 and box 1)- do they end at 5.0 and not 6.30pm? Please confirm. 4. Abbreviations (page 4, line 31) - please include abbreviations in brackets when these terms/names are introduced for the first time. 5. Missing data for 2 CCGs (page 6, line 43). This cannot be helped, but would it have been possible to compare the population characteristics of these with the remaining 5? It is a possible limitation. 6. Deprivation scores (page 7, line 17). These are important predictors of health needs. It is frustrating that individual patient scores were unavailable, but practice and CCG scores are available on PHE website (practice profiles). It is crude and not ideal, but would including practice IMD scores as a confounder in the multivariable analyses have been useful? 7. GGPS data (page 9, line 39). I am slightly unclear here- was the weighting done by the authors and not by the GPPS? The low
---

	response rate to GPPS is an issue, but they claim to have addressed this by their own weighting. 8. DNAs. This is an important area which may have benefitted from a bit more attention. The fact that nearly 1/5 of pre-booked GP appointments were not kept has substantial implications for service planning and resource allocation. I note that Sundays seem to have the lowest DNA rate (supplementary Figure 3), but did the authors identify any patient features (age gender) that predict DNA rates?
--	--

REVIEWER	Sam Watson University of Warwick, UK
REVIEW RETURNED	14-Jan-2019

GENERAL COMMENTS	This article is a descriptive analysis of a scheme to provide extended, out-of-hours primary care in the Greater Manchester area. The article describes appointment uptake and attendance as well as demographics of the patient population using the service. The results present useful information for future planning and analyses of these services. Comments  - I found the terminology confusing. In particular, the distinctions between appointments, booked appointments, used appointments, missed appointments etc. For example, in the results section the proportions of appointments booked and used is reported, but it is unclear what the denominator is each time: 'appointments used' could be interpreted as the proportion of possible appointment times or the proportion of booked appointments. Similarly in the Analysis section (page 7) 'The probability an appointment was booked' but it is unclear to me whether this means the outcome data were observation level (i.e. only booked appointments), booked v. walk-ins, or appointment slots/times which could be booked or not. Could I suggest changing the terminology to make this clearer? For example, 'consultation time' or 'slot'; 'booked appointment'; and 'attended, booked appointment' or 'attended consultation time', might make things easier to follow depending on what's appropriate. - It was unclear to me what the purpose or benefit of including the probit regressions is. The authors don't specify what they're trying to estimate and why the regression model is needed for it, but it seems as though they want to determine the proportion of consultation slots that were booked with appointments by day of the week, the proportion of booked appointments that were attended by day of the week, and the proportion of bookings by type and demographic etc. It's not made clear why the crude estimates are 'wrong' in this sense and why they might be "confounded" by month. These data are in essence the whole population of consultations and appointments. Indeed, the marginal differences in the tables are very similar to the actual crude differences. They do distract somewhat from the descriptive statistics, could the authors perhaps provide a stronger justification for using them and what they're estimating? Figures that explore the data could be more useful in the space. - Figure 4 is useful in that the comparison between 'core users' and 'extended hour users' is important. However, it is difficult to
---

	make comparisons with the different graphs in Figure 4 as: (a) the core users excludes under 18s; and (b) the pyramids are in absolute numbers rather than relative. It would be far easier to compare if they were on the same scale (relative) as say proportions of adult population, excluding the under 18s from the graph. I understand the bins are different, but if there's any way of overlapping the graphs this could also help, but could be difficult to interpret if it is too 'busy'. - This analysis and program fits in with the broader 'weekend effects' literature and out-of-hours care. In this literature Wednesday is often used as the comparator day to weekends as it is most representative of the week; Mondays are often the busiest days as they have the spillover from the weekend and patients who have waited to attend. This may (or may not) explain some of the Monday effects shown in the paper, but it could be useful to link to this broader literature. In addition, on this basis, it could be a little misleading to make the comparisons here with Monday as it is more like the weekend days than the weekday days at face value. Indeed, this relates to the first point above about the purpose of the comparisons. Minor comments - Page 5 line 54: 'whether there is a statistically significant difference...' why is this important to analysing differences in uptake? - Page 6 line 37: GPAF – acronym is undefined up to here I believe - Page 9 line 40: 'sampling weights to adjust...' what population was the sample adjusted to match? - Table 1: in the appointments column there is a (P=...) what does this mean? - Table 1: DNA's acronym undefined before this table, could define in table footnotes - Table 2 (and elsewhere): 'Pre-booked' could the authors define this term, i.e. does it mean the appointment was booked over 24 hours ago or on a previous calendar day or something else? - Results (e.g. page 11 lines 51- 60): could you define the denominators for the percentages, for example "weekday attendance was generally higher at 73.29%...." of what, consultation slots or booked appointments? - Table 13 line 19: The results may not have been statistically significant but this is of secondary importance to whether they were clinically or economically significant! The differences in Table S4 are of comparable magnitude to, say 'females v males pre-booking' (page 15, line 16) yet one is dismissed and the other deemed important only on the basis of 'statistical significance' Neither seem clinically or economically meaningful to me. - Page 15 lines 17-20: "A greater percentage... and 90 years over." The only comparisons made in Table 4 for which statistical significance is estimated is between each age group and 0-9 so
--	---

	one cannot make the comparison between, say 10-90 and 90+ on this basis. I would suggest either performing the test of interest or describing the differences. - Page 18 line 38: "...and same-day appointments (87% compared to 22% and 57%)." It's not clear what the three percentages refer to as the text only seems to refer to same-day and pre-booked appointments. - Figures: It may help the reader to put the acronyms in footnotes or the caption to facilitate interpretation of the figures. - Abstract: is it reasonable to suggest the participants are 1.2 million in number when no analysis is provided of this many people?!
--	--

REVIEWER	John Ford Cambridgeshire and Peterborough CCG, UK I have written a personal view article discussing the potentially negative impacts of weekend opening. See https://www.bmj.com/content/350/bmj.h1373
REVIEW RETURNED	16-Jan-2019

GENERAL COMMENTS	Thank you for inviting me to review this paper. It is a well written and robust study. It evaluates extended access in primary care across 13 centres in greater Manchester. I have the following major comments:  • My main issue is that the paper would benefit from being more balanced about the whether extended hours leads to meeting previously unmet need or supply-induced-demand. It may simply be that providing more appointments allows for the worried well to see a health professional. More health care is not always a good thing. For example, walk-in-centres provided more appointments but haven't appeared to improve care or reduce hospital admissions. Commissioners are now finding it challenging to decommission walkin centres because the public have got used to it. At present supply induced demand is only mentioned in the final "Future work" section. I suggest the following changes - I do not think the study provides any robust evidence about unmet need, therefore it should not be mentioned in the conclusions of the abstract. The implications section should discuss supply induced demand. • My second main issue is that there is no counterfactual. And while the study was not designed to have a counterfactual, the authors should mention this. It's unsurprising that if more appointments are provided people use them. The key question is should resource be invested in extended opening or core hours. Furthermore there is little discussion about the opportunity cost of extended hours. • My third main issue is that one key political driver behind extended opening is to reduce pressure on secondary care. This study does not provide any evidence on secondary care use which limits its potential impact. This need highlighted in the paper.
---

	I have the following minor comments:  • Some qualitative data would be useful and I note that the protocol mentions qualitative interviews. It would be useful for readers to hear about where they can access the qualitative findings. • In table 1 there are a few columns with the same heading. It would be easier to understand if the column headings were more specific. • The lack of deprivation data is a significant limitation and it's a shame that the reader is unable to see what the impact on inequalities would likely be. • I don't think reporting to two decimal places is required throughout. Perhaps the authors would consider reporting to only one decimal place for the proportions in the text to make it easier to read? • I note the authors state that females are less likely to pre-book with 0.66 less percentage points. Does this mean that the difference is 0.66% less compared to males? If so, this is not clinically significant although it might be statistically significant. • Should the analysis be clustered at practice level as well as CCG level. It may be that the behaviour of receptionists is patterned across different practices leading to a clustering effect. For example, anecdotal evidence suggests that patients are more likely to visit a hub at the weekend if the hub is located in their own practice.
--	--

REVIEWER	Paul Windrum University of Nottingham Business School
REVIEW RETURNED	22-Jan-2019

GENERAL COMMENTS	I welcome very much research in this topic area. It is of great relevance to current discussions and long term policy within primary care, and the NHS more widely. There are two aspects of the paper. One is a descriptive analysis of the take-up of extended hours services offered to patients. Overall, the analysis is fine. The interpretation is open to question. First, there is a statement that this meets a previously unmet need. This is not substantiated in the paper. It requires closer consideration. The service provided is non-emergency care. Emergency cases are selected out of the population and directed to emergency services. Have alternative sources of provision not been used for these 'urgent' but 'non-emergency' patients in the past? One can argue that the A&E Department is not the best place for patients to end up, but this is not the same as stating there is or was an unmet need. I would also be more cautious of interpreting an average 65% take-up of available appointments as a success. To an economist, that sounds like an inefficiency of 35%. These are scarce resources that could be deployed elsewhere. I would have welcomed a more detailed analysis here of the initial (ex ante) expectations of patient demand when first planning these services (which are presumably reflected in the size of the service as well as the type of service offered, and the number of appointments made available), and the actual (ex post) demand of patients as reflected by attendance.
--

	The second part of the paper I find much more problematic. This is a statistical analysis in which used appointments in the extended hours service are compared to used appointments in core hours. Best research practice is to create a matched sample where individuals are matched on personal characteristics such as age, socio-economic status etc . There are various procedures available for doing this, however this paper does not use this approach. Instead, information for the comparison/control group is drawn national patient survey data that does not provide matching information on personal characteristics. The national patient survey data that is used is also from a previous time period.
--	--

REVIEWER	Irene Stratton Gloucestershire Retinal Research Group UK
REVIEW RETURNED	25-Feb-2019

GENERAL COMMENTS	My first comment on this paper is that these practices are nothing like the one I am registered with where a request for an appointment may come up with one 4 or 5 weeks away - even an appointment with a nurse for a chronic health condition check could be 6 weeks hence. So perhaps generalisability could be discussed in the light of this? From the figures only 50% of appointments were used in January.... that wouldn't be the case in GP practices in Oxfordshire! Does "nurse" include midwives? Do you have any information on appointments that were made but that were then cancelled by the patient? (can't find a mention of the word "cancelled"). How were patients in a GP practice made aware that their practice would be offering extended hours? If they did not know they might just go to A&E without calling the GP on a Saturday or Sunday. Did anyone notice that CCG4 had least good data capture before the end of the study? This would, for me, be one of the major findings. I have looked using Word search function and can't see any mention of "data quality" or "completeness". Figure 3 loses the information that Sunday is next to Monday. This could be seen using a plot as shown here: https://boraberan.wordpress.com/2014/03/30/creating-coxcomb-charts-in-tableau/ The most famous example of this is from Florence Nightingale of course.... Figure 4. Oh dear. The y axes are not the same. Same problem with reporting in S8. Could you think of a way of putting both sets of data on the same plot? Is it possible to differentiate between routine appointments for chronic health monitoring - diabetes for example - and new health problems?
---

	Could you not use a deprivation indicator for the GP practice? "DNA appointments were more likely to be pre-booked than attended appointments (18.85 percentage points, 95%CI: 16.88 to 20.83)). " Would this not be easier to understand if it were to be expressed as "patients were more likely to turn up to appointments they had made that day than appointments which had been pre-booked"? Perhaps the biggest problem is that we have no information here on the population on the books of the GP practices. "The most obvious difference was found in females aged 20-29 who accounted for almost a quarter of extended access appointments users (aged 20+). " So this might be because there were a lot of young women registered with the practices in the study (students for example)? My overall impression of this paper is that the work has been designed by health economists and carried out and analysed and reported on by health economists, rather than by a team which might include a GP and a primary care practice administrator. There are a lot of important issues not mentioned - the background population demographics (age distribution), whether appointments are for monitoring chronic health problems or maybe pregnant women visiting midwives, advertising of extended intervals, whether text reminders are used.
--	---

REVIEWER	Helen Purtill University of Limerick, Ireland
REVIEW RETURNED	25-Feb-2019

GENERAL COMMENTS	This study aims to examine the uptake of extended primary care service within 13 centres in Greater Manchester Area in England in 2016. The authors are to be commended for an interesting study, a well written manuscript and detailed descriptions of the data through the tables, graphs and appendices. However, I've strong concerns about the statistical models and the conclusions drawn from the analyses, as outlined below. Abstract: Objectives - The first objective is clear. The secondary objective looks to examine patient differences across the appointment types and to compare the demographic profile of the extended primary care service with that of core hour users.  - the first objective relates to the appointments - the second objective relates to the patients that make the appointments Participants - 1,261,326 are not the participants. This is the population size with access to the 5 CCGs. In this study there is a sample of 42,472 appointments. There is only have information on those who make appointments (not those who don't make the appointments). Information is not given in the study on how many participants made the 32041 appointments over the four CCGs analysed for the patient demographic analysis - it would be good to have a measure of the exact number of patients made the 32041 appointments and thus have an idea of the number of patients making repeat visits.
---

In the results section of the abstract it is stated that "Uptake increased over the 2016 calendar year" - this wasn't the case. In table 3, the adjusted marginal percentages for "Appointment Used" for August is lower than July and June. The adjusted marginal percentage for December is similar to April - there is no clear pattern of an increase.

Line 25 on Page 2: "Pre-booked appointment participants were less likely to be female" - this is from the Probit analysis presented in Table 4. But in this regression model there are a mixture of variables that measure patient characteristics (eg gender, age) and variables that measure time (day of week, month of year) and variables that measure the appointment status. If you'd like to determine whether female patients are more likely to pre-book an appointment compared to male patients, you should not control for day, or month, or appointment type - these are not measured on the patient. The analysis as presented in Table 4 (using the appointment as the unit of analysis) cannot be interpreted as if the patient was the unit of analysis.

Even if the analysis was interpreted correctly a difference of 0.66 percent points is an extremely small effect and unlikely to be meaningful in the context of the study.

Line 29 & 30 of pg 2: Uptake improved over time, for all week days. This statement is too strong for the analysis in Table 3. In table 3, the adjusted marginal percentages for "Appointment Used" for August is lower than July and June. The adjusted marginal percentage for December is similar to April - there is no clear pattern of an increase.

Line 5 of Pg 8 "Probit models were chosen as these give the absolute variation in uptake". I understand what is meant but I'd prefer to see something like "Probit models were chosen as these estimate differences in the probability of uptake across categories of the predictor variables". The word "variation" is more usually seen in the context of quantitative outcome variables (eg The explained variation in by a fitted linear regression model).

Analysis of Uptake:

It is not clear in this section if "uptake" refers to the appointment being booked or if it refers to the appointment being booked and used. Please give very specific definition for each case and use that definition throughout the analysis.

1. CCG 5 is chosen as the base centre but doesn't have any weekday appointments (see Table S1) and so the model will have empty cells for day of week Mon-Fri for CCG5. A probit model should not have empty cells.

2. January is compared as the base month but looking at the start dates it is not surprising at all that the January numbers are so low (Table S1) - many hubs don't start until the middle, end or after January. February would seem a more reasonable comparison group for the probability of appointments booked and probability of appointment used.

3. The title of Table 3 "Rates and probability models for appointment use" doesn't make sense. Are all the figures in table 3 estimated by the Probit models? Or are the % booked summary statistics?

4. CCG is used to control for CCG area and standard errors are adjusted for patients being clustered within these areas. But there could be a hub effect. Hub could be included as a random effect in the analysis instead of the fixed effect of CCG.

Lines 15 to 20 on page 13: The conclusions given from a non-significant result are incorrect. If a statistical test is non-significant you cannot imply there is equivalence - only that you don't have evidence in your data of the effect you were testing for. To test for an interaction of Sunday by month (ie if the effect of Sunday changed overtime) it would be sufficient to include an indicator variable "Sunday" coded as 1 = yes and 0 = no, the dummy variables indicating month, the CCG (or Hub) variables and Sunday by month interaction variables.

Patient Demographic Analysis:

The research question surrounds the patients, and examining patient demographics associated with Nurse V GP and Pre-booked V Same day appointments. But the unit of analysis presented in the Probit regression (Table 4) is an appointment booked, not a patient. To run and interpret the analysis with respect to patient demographics the analysis needs to be run using the patient as the unit of the analysis (patient variables are gender, age and CCG or hub attended). You need to consider if there are repeat patients. If there are not many repeat patients you may assume independence of observations but this needs to be clearly stated as an assumption and you need to state it as a limitation (ie that the same person may be included in the analysis twice).

An analysis of the patient demographics vs booked appointments and used appointments is interesting - from the descriptive Table S1 it appears that more females use the service compared to males, as also seen in the graph on the left of Figure 4, which shows the differences in the shape of the age profile of male and female patients. But, maybe the population is younger with more females? A descriptive table of Nurse Vs GP and Pre-booked Vs Same day across the different variables would be interesting too.

Would it be of interest to examine factors associated with DNA appointments (ie whether or not the appointment was used)?

Eg Are patient demographics predictive of DNA? Or you could use the appointment as the unit of analysis to examine if any appointment factors associated with DNA? A regression model could answer: are appointments made for GPs or Nurses more likely to be DNA? Are appointments made on same day v pre-booked more likely to be DNA? etc, are appointments made by a female, made on different days of the week more likely to be DNA.. etc..

Comparisons of extended access appointment to core hour users: Yes, I'd agree that caution needs to be exercised because the age bands do not align in the datasets. You are also assuming that the gender and age profile for patients from in England (GPPS survey) is representative of the patient profile for core use in the Greater Manchester area - Is this the case? It should be included in a limitation that this may not be the case.

	Discussion: Line 56, 57 - should be "..... who accounted for almost a quarter of all female extended access...." A table of descriptive statistics for GP V Nurse and Pre booked Vs Same Day across the patient demographics would be informative.
--	--

VERSION 1 – AUTHOR RESPONSE

Reviewer: 1

Reviewer Name: Professor Shona J. Kelly

Institution and Country: Sheffield Hallam University, United Kingdom

Please state any competing interests or state 'None declared': none declared

Please leave your comments for the authors below

Re: bmjopen-2018-028138

Thank you for the opportunity to review this paper that investigated the use of extended access to primary care services in Greater Manchester. The findings are extremely interesting but some work is needed to make the methodology accessible for a non-statistical audience.

1. The biggest issue the use of a probit model. Why probit? This type of regression is very uncommon in public health/epidemiology. My statistics is at a moderately advanced level and my understanding from the explanation offered by my statistician is that it is used when there are a lot of empty cells. I presume that you had the same problem as we did on a similar set of data and had to deal with a lot of missing data and some appointment slots (on Sundays) that were not taken up? The explanation in the supplementary file is not understandable except by a statistician. You need to find a way to explain this type of regression. Most of the readers of this paper will be familiar with logistic regression so a comparison with it would be a good place to start your explanation of a probit model. You also need to explain how to interpret the tables. For example in Table 3, the "appointment booked" column - how does one interpret the values - the title says "rates", is the usual number of appointments per xxxx patients?

R1.1 Author response: We have expanded our rationale for the choice of the probit model in the methods section of the manuscript (Page 7, paragraph 2). We have also provided an example to aid in interpretation (Page 11, paragraph 2). To help reassure readers that the results of the study are robust across model specifications we have also added Supplementary Table S5 which presents estimates from both probit and logit regressions.

2. The quote in the introduction should have the reference at the end removed as it is confusing

R1.2 Author response: We have removed the reference as suggested (Page 3, paragraph 2).

3. Define acronyms the first time they are used - e.g. 'GM'

R1.3 Author response: We have defined acronyms that had been overlooked (GM (Page 3, paragraph 2), PMCF (Page 5, paragraph 3), GPAF (Page 5, paragraph 3)). Thank you for noticing this.

4. I would have put summary table 1 in the text and put your table 1 in an appendix. Knowing which are nurse or doctor appointments is important for interpreting the findings.

R1.4 Author response: We have not made the suggested change here. This is because the variation in services delivered can be seen in Table 2 and Table 1 gives useful information on volumes of appointments available which we feel is key to the context of the paper. We have attempted to reduce the content in Tables 1 and 2 (by removing the CCG-hub activity breakdowns) which may better bring these points to light in the manuscript.

5. you need to explain in methods that the data on core hour practice use is patient satisfaction data and it is pretty much useless

R1.5 Author response: We have added a comment clarifying that the GPPS is a patient satisfaction survey and that the aim of the survey is not to capture volumes of general practice activity (Page 8, paragraph 3). However, we have also argued that this is the most appropriate dataset available given the lack of alternatives.

6. Were these practice nurse appointments or advanced nurse practitioner appointments?

R1.6 Author response: The appointments were with a nurse practitioner. We have clarified this in the manuscript (Page 6, paragraph 1; Page 8, paragraph 1). Please note nurse practitioners are also detailed in Box 1 where we describe the service.

7. I'm not sure all the detail on the Hubs and CCGs is relevant to the general audience reading this paper. I'd remove all of that as people can read your full online report which you can reference.

R1.7 Author response: We have reduced the detail in Tables 1 and 2 by removing the hub breakdowns of activity (see R1.4 Author response). Beyond this we have retained a focus on CCGs, this is because the variation in CCGs is an important aspect of the study, highlighting the variation in approaches taken.

8. A comment in the discussion about why the CCGs could get away without providing data would be illuminating.

R1.8 Author response: We have added a discussion of the limits of the research team in enforcing complete data capture in the discussion (Page 17, paragraph 1).

9. The use of white text on blue in figure 1 is too hard to read.

R1.9 Author response: We have changed the colour scheme for Figure 1 to a black and white theme which hopefully helps with regards to readability.

10. Some of the figures need more footnotes or information in the titles. e.g. Figure 2 needs n(%)

R1.10 Author response: We have edited the caption for Figures 2 and 3 to clarify volume and percentages are presented in these Figures. We have also added footnotes to all Figures (Page 23, 'Figures' section).

11. Figure 4 needs a note about why the under 18's are missing in the right hand chart. They would also be more comparable if the two charts were scaled the same. I think the left-hand chart is key for the discussion.

R1.11 Author response: We have rearranged this section of the manuscript to aid in clarity. In particular, we have presented the total population of extended access users separately to the GPPS comparison (Figure 4) and have also provided the registered population to aid in comparisons with the general population in the area. We have separately presented the extended access users aged 20+ with the GPPS users (Figure 5). We have focussed attention on the total population of extended access users in the main manuscript by switching Supplementary Table S8 with Table 5.

12. It is not usual to include the contract as an appendix. You could refer the reader to an online version. It is also confusing because this paper doesn't address all of the outcomes listed in the contract.

R1.13 Author response: Please note that in accordance with BMJ Open requirements to include study protocols in the submission process we have retained Supplementary Text S1 in the manuscript.

13. Some general statements about data accessibility and quality would be useful. Why isn't data easily available? Why can't the CCGs provide the postcode derived deprivation score? Why don't we have better demographics?

R1.14 Author response: We have further explained the process by which data arrived and the limits the research team had in terms of data quality and coverage (Page 6, paragraph 1). Here we have also added a sentence to highlight that patient confidentiality restricted the range and depth of patient characteristics captured.

Reviewer: 2

Reviewer Name: Louis Levene

Institution and Country: University of Leicester, United Kingdom

Please state any competing interests or state 'None declared': None declared

Please leave your comments for the authors below

General points:

Overall, this paper covers an important and interesting topic. The authors have set out a good research question and used sound methodology to answer it (although I could not identify any issues, it may help to have a statistician double-check it as I am not a statistical expert). They have interpreted their results judiciously and have considered the key points and implications.

Specific points (apologies for seeming pedantic):

1. Dates covered in 2016 (page 6, line 57). Assuming it is 1 Jan to 31 Dec- just need to make this clear.

R2.1 Author response: We have clarified the months covered (January to December) (Page 6, paragraph 1).

2. Outcomes- primary vs. secondary. This is made clear in the abstract and introduction, but not at the start of the discussion (principal findings).

R2.2 Author response: We have expanded the discussion to separate primary from secondary analyses more clearly (Page 16, paragraph 1).

3. Core hours (page 4, line 17 and box 1)- do they end at 5.0 and not 6.30pm? Please confirm.

R2.3 Author response: We have changed the closing time to 6:30pm to reflect that core hour opening times vary in general practice, thank you for noticing this (Page 3, paragraph 2 and Box 1).

4. Abbreviations (page 4, line 31) - please include abbreviations in brackets when these terms/names are introduced for the first time.

R2.4 Author response: Please see the R1.3 Author response which covers this comment.

5. Missing data for 2 CCGs (page 6, line 43). This cannot be helped, but would it have been possible to compare the population characteristics of these with the remaining 5? It is a possible limitation.

R2.5 Author response: Thank you for this suggestion. Please note that of the two CCGs one was not providing an extending access service in 2016 and one did not provide any data (Page 5, paragraph 3). We consider the underlying issue being raised is one of generalisability and the population characteristics are just one factor of the more broader issue of service generalisability - in particular, our results may not transfer across to schemes that have alternative specifications for an extended access service. We have expanded our limitations to further develop this argument (Page 17, paragraph 1).

6. Deprivation scores (page 7, line 17). These are important predictors of health needs. It is frustrating that individual patient scores were unavailable, but practice and CCG scores are available on PHE

website (practice profiles). It is crude and not ideal, but would including practice IMD scores as a confounder in the multivariable analyses have been useful?

R2.6 Author response: Many thanks to the reviewer for considering alternative approaches to deal with missing deprivation data. Whilst we do appreciate the suggestion, we have kept a distance from including practice measures beyond the clustering of standard errors. This is because they may be a poor proxy for patient circumstance as the reviewer suggests.

7. GGPS data (page 9, line 39). I am slightly unclear here- was the weighting done by the authors and not by the GPPS? The low response rate to GPPS is an issue, but they claim to have addressed this by their own weighting.

R2.7 Author response: We have clarified that the weights used were included in the GPPS data (Page 9, paragraph 1).

8. DNAs. This is an important area which may have benefitted from a bit more attention. The fact that nearly 1/5 of pre-booked GP appointments were not kept has substantial implications for service planning and resource allocation. I note that Sundays seem to have the lowest DNA rate (supplementary Figure 3), but did the authors identify any patient features (age gender) that predict DNA rates?

R2.8 Author response: Prior to submission, earlier drafts of the manuscript had included models of DNA. To reduce word count we excluded these on the basis that DNA was a predictor variable in the GP/nurse and pre-booked/same-day analyses and some inference could be made about DNAs from these models. However, this did indeed come at a cost of losing a demographic context for DNA. We have reintroduced the DNA analyses in the updated manuscript given this provides some useful additional content relating to demographics (Table 4, Page 13).

Reviewer: 3

Reviewer Name: Sam Watson

Institution and Country: University of Warwick, UK

Please state any competing interests or state 'None declared': None declared

Please leave your comments for the authors below

This article is a descriptive analysis of a scheme to provide extended, out-of-hours primary care in the Greater Manchester area. The article describes appointment uptake and attendance as well as demographics of the patient population using the service. The results present useful information for future planning and analyses of these services.

Comments

- I found the terminology confusing. In particular, the distinctions between appointments, booked appointments, used appointments, missed appointments etc. For example, in the results section the proportions of appointments booked and used is reported, but it is unclear what the denominator is each time: 'appointments used' could be interpreted as the proportion of possible appointment times or the proportion of booked appointments. Similarly in the Analysis section (page 7) 'The probability an appointment was booked' but it is unclear to me whether this means the outcome data were observation level (i.e. only booked appointments), booked v. walk-ins, or appointment slots/times which could be booked or not. Could I suggest changing the terminology to make this clearer? For example, 'consultation time' or 'slot'; 'booked appointment'; and 'attended, booked appointment' or 'attended consultation time', might make things easier to follow depending on what's appropriate.

R3.1 Author response: Thank you for raising this confusion. Throughout the manuscript we have clarified where the analyses concern appointments booked (used and not attended), appointments booked and used, appointments booked and not attended (DNAs), and appointments not booked (the

first sentence to each sub-section of the results). To aid clarity we have also refrained from discussing uptake beyond the primary set of analyses as this could mean appointments booked or appointments booked and used. We have also made changes throughout the manuscript to ensure consistency in terminology (where possible by using 'used' rather than booked and attended or uptake, see Table 1 column headings for example).

- It was unclear to me what the purpose or benefit of including the probit regressions is. The authors don't specify what they're trying to estimate and why the regression model is needed for it, but it seems as though they want to determine the proportion of consultation slots that were booked with appointments by day of the week, the proportion of booked appointments that were attended by day of the week, and the proportion of bookings by type and demographic etc. It's not made clear why the crude estimates are 'wrong' in this sense and why they might be "confounded" by month. These data are in essence the whole population of consultations and appointments. Indeed, the marginal differences in the tables are very similar to the actual crude differences. They do distract somewhat from the descriptive statistics, could the authors perhaps provide a stronger justification for using them and what they're estimating? Figures that explore the data could be more useful in the space.

R3.2 Author response: We have expanded the methods section to further justify the use of regression models (Page 7, paragraph 2). This is for two main reasons - the potential uncertainty in the data and the ability to adjust for CCG effects which we feel are important given the variation in schemes across CCGs.

- Figure 4 is useful in that the comparison between 'core users' and 'extended hour users' is important. However, it is difficult to make comparisons with the different graphs in Figure 4 as: (a) the core users excludes under 18s; and (b) the pyramids are in absolute numbers rather than relative. It would be far easier to compare if they were on the same scale (relative) as say proportions of adult population, excluding the under 18s from the graph. I understand the bins are different, but if there's any way of overlapping the graphs this could also help, but could be difficult to interpret if it is too 'busy'.

R3.4 Author response: Thank you for this helpful suggestion. We have adapted the figure accordingly by presenting the percentage shares which give the same scale over the datasets (Figure 5). We feel these changes help highlight the differences between the two groups clearly.

- This analysis and program fits in with the broader 'weekend effects' literature and out-of-hours care. In this literature Wednesday is often used as the comparator day to weekends as it is most representative of the week; Mondays are often the busiest days as they have the spillover from the weekend and patients who have waited to attend. This may (or may not) explain some of the Monday effects shown in the paper, but it could be useful to link to this broader literature. In addition, on this basis, it could be a little misleading to make the comparisons here with Monday as it is more like the weekend days than the weekday days at face value. Indeed, this relates to the first point above about the purpose of the comparisons.

R3.5 Author response: We thank the reviewer for their concern and thoughts here, but we argue that the choice of base category would not lead to alternative findings of the study. We have maintained Monday as the base category to ease in interpretation as Monday may be seen as the start of the week. To avoid complicating the motivation of the study and potentially confusing the reader we have not incorporated a discussion of the weekend effects literature. Moreover, it should be noted that Wednesday is chosen as the base category in the weekend effect literature as it is the quietest day in the hospital and therefore the one when processes should be running more smoothly (as suggested). There is no reason why a similar logic could apply to the extended access service.

Minor comments

- Page 5 line 54: 'whether there is a statistically significant difference...' why is this important to analysing differences in uptake?

R3.6 Author response: We consider the issue of testing for statistical differences to be important in the study given the nature of the data. The ability to incorporate uncertainty in the data is important, particularly where data may not be complete. However, we appreciate that this statement requires justification and expansion in the introduction. We consider the statement to be superfluous to the argument and as this is detailed elsewhere in the manuscript (Page 7, paragraph 2), we have omitted this from the introduction (Page 4, paragraph 1).

- Page 6 line 37: GPAF – acronym is undefined up to here I believe

R3.7 Author response: Please see the R1.3 Author response which covers this comment.

- Page 9 line 40: 'sampling weights to adjust...' what population was the sample adjusted to match?

R3.8 Author response: We have clarified the weights were included in the GPPS and generated to ensure age and gender generalisability (Page 9, paragraph 1).

- Table 1: in the appointments column there is a (P=...) what does this mean?

R3.9 Author response: We have clarified this is the population in that area (Page 10, Table 1).

- Table 1: DNA's acronym undefined before this table, could define in table footnotes

R3.10 Author response: We have now defined DNA in the caption for Table 1 (Page 10, Table 1).

- Table 2 (and elsewhere): 'Pre-booked' could the authors define this term, i.e. does it mean the appointment was booked over 24 hours ago or on a previous calendar day or something else?

R3.11 Author response: We have clarified that pre-booked means the appointment was booked on a previous calendar date in the text (Page 6, paragraph 1) and Table 2 (Page 11) and Table 4 (Page 13).

- Results (e.g. page 11 lines 51- 60): could you define the denominators for the percentages, for example "weekday attendance was generally higher at 73.29%...." of what, consultation slots or booked appointments?

R3.12 Author response: We have expanded the description in the results section to clarify this was the percentage of appointments booked and used (Page 11, paragraph 1).

- Table 13 line 19: The results may not have been statistically significant but this is of secondary importance to whether they were clinically or economically significant! The differences in Table S4 are of comparable magnitude to, say 'females v males pre-booking' (page 15, line 16) yet one is dismissed and the other deemed important only on the basis of 'statistical significance' Neither seem clinically or economically meaningful to me.

R3.13 Author response: We thank the reviewer for highlighting this. Under the new set of estimates we note where a significant effect is of a small size (see for example, Page 11, paragraph 2 concerning month differences).

- Page 15 lines 17-20: "A greater percentage... and 90 years over." The only comparisons made in Table 4 for which statistical significance is estimated is between each age group and 0-9 so one cannot make the comparison between, say 10-90 and 90+ on this basis. I would suggest either performing the test of interest or describing the differences.

R3.14 Author response: Given the estimated effect was insignificant for age group 90+ this would suggest this group had similar percentages to age group 0-9 (the base group). However, we have

removed the reference to age group 90+ as we appreciate that age group 90+ has not been formally tested against the other age group effects (Page 14, paragraph 1).

- Page 18 line 38: "...and same-day appointments (87% compared to 22% and 57%)." It's not clear what the three percentages refer to as the text only seems to refer to same-day and pre-booked appointments.

R3.15 Author response: We have clarified that this was comparisons of three schemes that provided same-day only or same-day and pre-bookable appointments (Page 19, paragraph 2).

- Figures: It may help the reader to put the acronyms in footnotes or the caption to facilitate interpretation of the figures.

R3.16 Author response: We have expanded the footnotes to Figures 2 to 4 to enable the Figures to be self-standing (Page 23, Figures).

- Abstract: is it reasonable to suggest the participants are 1.2 million in number when no analysis is provided of this many people?!

R3.17 Author response: We have edited the participants section of the abstract to clarify the focus on appointments and that the 1.2m refers to a population who could book on to the service.

Reviewer: 4

Reviewer Name: John Ford

Institution and Country: Cambridgeshire and Peterborough CCG, UK

Please state any competing interests or state 'None declared': I have written a personal view article discussing the potentially negative impacts of weekend opening.

See <https://www.bmj.com/content/350/bmj.h1373>

Please leave your comments for the authors below

Thank you for inviting me to review this paper. It is a well written and robust study. It evaluates extended access in primary care across 13 centres in greater Manchester.

I have the following major comments:

- My main issue is that the paper would benefit from being more balanced about the whether extended hours leads to meeting previously unmet need or supply-induced-demand. It may simply be that providing more appointments allows for the worried well to see a health professional. More health care is not always a good thing. For example, walk-in-centres provided more appointments but haven't appeared to improve care or reduce hospital admissions. Commissioners are now finding it challenging to decommission walkin centres because the public have got used to it. At present supply induced demand is only mentioned in the final "Future work" section. I suggest the following changes - I do not think the study provides any robust evidence about unmet need, therefore it should not be mentioned in the conclusions of the abstract. The implications section should discuss supply induced demand.

R4.1 Author response: We have expanded the discussion concerning unmet need to incorporate inappropriately met need and the potential for supply induced demand (Page 17, paragraph 2; Page 20, paragraph 1). Whilst we consider supply induced demand to be unlikely given the findings of the Healthwatch Manchester study (reference 16) that awareness is a key concern, we appreciate we cannot rule out this without further evidence of the reasons for attendance and services delivered in an appointment. We have removed mention of unmet need in the abstract.

- My second main issue is that there is no counterfactual. And while the study was not designed to have a counterfactual, the authors should mention this. It's unsurprising that if more appointments are provided people use them. The key question is should resource be invested in

extended opening or core hours. Furthermore there is little discussion about the opportunity cost of extended hours.

R4.2 Author response: We have expanded our discussion relating to cost-effectiveness to mention that we do not compare to a counterfactual and to incorporate more explicitly the rationale behind such evaluations which has better highlighted the issue of opportunity cost (Page 18, paragraph 3).

- My third main issue is that one key political driver behind extended opening is to reduce pressure on secondary care. This study does not provide any evidence on secondary care use which limits its potential impact. This need highlighted in the paper.

R4.3 Author response: Since submitting the manuscript our final report has been released (though dated March 2017 there were delays in publication). We now provide a link to the report and highlight that this study concerns the activity assessment component of the report, to include secondary care impacts would take a substantial amount of content and we would argue would best be submitted as a separate manuscript in itself.

I have the following minor comments:

- Some qualitative data would be useful and I note that the protocol mentions qualitative interviews. It would be useful for readers to hear about where they can access the qualitative findings.

R4.4 Author response: Please see the above response relating to the release of the final report and linkage provided.

- In table 1 there are a few columns with the same heading. It would be easier to understand if the column headings were more specific.

R4.5 Author response: We have altered Table 1 in response to earlier reviewer comments and this expanded the column titles to better explain the contents (Page 11, Table 1).

- The lack of deprivation data is a significant limitation and it's a shame that the reader is unable to see what the impact on inequalities would likely be.

R4.6 Author response: We agree with the reviewer that this is a shame. We hope to inform this in the future. This is something we are working on for a different pilot scheme that is currently in progress.

- I don't think reporting to two decimal places is required throughout. Perhaps the authors would consider reporting to only one decimal place for the proportions in the text to make it easier to read?

R4.7 Author response: We have retained the use of two decimal places to ensure consistency with the BMJ style.

- I note the authors state that females are less likely to pre-book with 0.66 less percentage points. Does this mean that the difference is 0.66% less compared to males? If so, this is not clinically significant although it might be statistically significant.

R4.8 Author response: In line with the comment by Reviewer 3 (R3.13), we have ensured we highlight where statistically significant estimates are small in size.

- Should the analysis be clustered at practice level as well as CCG level. It may be that the behaviour of receptionists is patterned across different practices leading to a clustering effect. For example, anecdotal evidence suggests that patients are more likely to visit a hub at the weekend if the hub is located in their own practice.

R4.9 Author response: Thank you for this suggestion. For the appointments booked analyses we now cluster on the patient booking the appointments GP practice (Page 8, paragraph 2), this does have implications on the sample size as 7.43% of appointments booked have a missing GP practice code (Supplementary Table S6). Nonetheless, for statistical inference the reviewer is right to raise this as a

suggestion and we agree it is important. The impact of this change is that all subsequent analyses on appointments booked is now on a smaller sample. We now find missing data is associated with week days and calendar month (Page 13, paragraph 1; Supplementary Table S7). We have expanded the limitations section to incorporate the limitations of this approach (Page 17, paragraph 1).

Reviewer: 5

Reviewer Name: Paul Windrum

Institution and Country: University of Nottingham Business School

Please state any competing interests or state 'None declared': None declared

Please leave your comments for the authors below

I welcome very much research in this topic area. It is of great relevance to current discussions and long term policy within primary care, and the NHS more widely.

There are two aspects of the paper. One is a descriptive analysis of the take-up of extended hours services offered to patients. Overall, the analysis is fine. The interpretation is open to question. First, there is a statement that this meets a previously unmet need. This is not substantiated in the paper. It requires closer consideration. The service provided is non-emergency care. Emergency cases are selected out of the population and directed to emergency services. Have alternative sources of provision not been used for these 'urgent' but 'non-emergency' patients in the past? One can argue that the A&E Department is not the best place for patients to end up, but this is not the same as stating there is or was an unmet need.

R5.1 Author response: Thank you for raising this. We agree that patients may have used other services in the healthcare system in the absence of an ability to make an appointment. In this sense the patient's need may be met. We do, however, argue that this is inappropriately met need (a response caused by an inability to obtain care in general practice) and have added this additional context to the implications section of the manuscript (Page 17, paragraph 2).

I would also be more cautious of interpreting an average 65% take-up of available appointments as a success. To an economist, that sounds like an inefficiency of 35%. These are scarce resources that could be deployed elsewhere. I would have welcomed a more detailed analysis here of the initial (ex ante) expectations of patient demand when first planning these services (which are presumably reflected in the size of the service as well as the type of service offered, and the number of appointments made available), and the actual (ex post) demand of patients as reflected by attendance.

R5.2 Author response: We had discussed ways to improve uptake in the implications section of the manuscript but the reviewer is right to note that this could be more significantly raised as an issue. We have therefore expanded the discussion of cost-effectiveness with poor uptake a focal point of point three (Page 18, paragraph 2). We have added a reference to the (now released) evaluation report that includes qualitative assessments of how the service was implemented (Page 5, paragraph 3).

The second part of the paper I find much more problematic. This is a statistical analysis in which used appointments in the extended hours service are compared to used appointments in core hours. Best research practice is to create a matched sample where individuals are matched on personal characteristics such as age, socio-economic status etc .

There are various procedures available for doing this, however this paper does not use this approach. Instead, information for the comparison/control group is drawn national patient survey data that does not provide matching information on personal characteristics. The national patient survey data that is used is also from a previous time period.

R5.3 Author response: The aim of the comparison to core hours is to see whether the patients are similar or not in terms of age and/or gender or not. For this reason matching would not be appropriate

as it is the variation in demography that we are measuring. Please also note that the earlier time point is used to ensure responses in the GPPS are not potentially biased by the use of extended access service (see Page 9, paragraph 1).

Reviewer: 6

Reviewer Name: Irene Stratton

Institution and Country: Gloucestershire Retinal Research Group, UK

Please state any competing interests or state 'None declared': None declared

Please leave your comments for the authors below

My first comment on this paper is that these practices are nothing like the one I am registered with where a request for an appointment may come up with one 4 or 5 weeks away - even an appointment with a nurse for a chronic health condition check could be 6 weeks hence. So perhaps generalisability could be discussed in the light of this? From the figures only 50% of appointments were used in January.... that wouldn't be the case in GP practices in Oxfordshire!

R6.1 Author response: Please note the study concerns extended access appointments, these are delivered outside of core hours (please see Box 1 on Page 3 for a description of the service). The reviewer inadvertently raises a point that was not very clear in the manuscript – that of patient awareness which, alongside practice (receptionist) buy-in, may impact on the use of extended access appointments. We have expanded the reasons behind spare capacity accordingly (Page 18, paragraph 1).

Does "nurse" include midwives?

R6.2 Author response: Please see response to R1.6.

Do you have any information on appointments that were made but that were then cancelled by the patient? (can't find a mention of the word "cancelled").

R6.3 Author response: Unfortunately we do not have the ability in the data to identify whether an appointment was cancelled. We have added this as a limitation (Page 17, paragraph 1).

How were patients in a GP practice made aware that their practice would be offering extended hours? If they did not know they might just go to A&E without calling the GP on a Saturday or Sunday.

R6.4 Author response: This is a good point, as mentioned in R6.1 we had raised the issue of patient awareness and the need to understand this in the implications section of the original manuscript although it was perhaps not very clear. We have modified the text in the implications section of the manuscript to better highlight this issue (Page 18, paragraph 1).

Did anyone notice that CCG4 had least good data capture before the end of the study? This would, for me, be one of the major findings. I have looked using Word search function and can't see any mention of "data quality" or "completeness".

R6.5 Author response: The research team did feed back any issues with data as it arose. However, it was up to the CCG to act on this information. We have added a discussion of the process of obtaining the data (Page 6, paragraph 1) and data quality (Page 17, paragraph 1).

Figure 3 loses the information that Sunday is next to Monday. This could be seen using a plot as shown here:

<https://boraberan.wordpress.com/2014/03/30/creating-coxcomb-charts-in-tableau/>

The most famous example of this is from Florence Nightingale of course....

R6.6 Author response: Thank you for this suggestion. Whilst we agree that presenting the day of week graph under such an approach is attractive, we feel the graph in its current state is more

consistent with the style of the calendar month graph and is perhaps more clearer to understand, and of the type expected, for the reader.

Figure 4. Oh dear. The y axes are not the same. Same problem with reporting in S8. Could you think of a way of putting both sets of data on the same plot?

R6.7 Author response: Please refer to the earlier response to Reviewer 3 (R3.4).

Is it possible to differentiate between routine appointments for chronic health monitoring - diabetes for example - and new health problems?

R6.8 Author response: Unfortunately we do not have access to the reason for the appointment or outcomes of the appointments to enable us to consider this useful suggestion. We do now make mention of the need for this information (Page 20, paragraph 1).

Could you not use a deprivation indicator for the GP practice?

R6.9 Author response: Please see our response to reviewer 2 (R2.6).

"DNA appointments were more likely to be pre-booked than attended appointments (18.85 percentage points, 95%CI: 16.88 to 20.83)). "

Would this not be easier to understand if it were to be expressed as "patients were more likely to turn up to appointments they had made that day than appointments which had been pre-booked"?

R6.10 Author response: We have taken the reviewers helpful suggestion (thank you) and edited the relevant sentence (Page 14, paragraph 1).

Perhaps the biggest problem is that we have no information here on the population on the books of the GP practices.

"The most obvious difference was found in females aged 20-29 who accounted for almost a quarter of extended access appointments users (aged 20+)." So this might be because there were a lot of young women registered with the practices in the study (students for example)?

R6.11 Author response: We now include the population registered with the practices in the area in Figures 4. Note however, that should the population have a lot of younger women then this should be reflected in core hour use too.

My overall impression of this paper is that the work has been designed by health economists and carried out and analysed and reported on by health economists, rather than by a team which might include a GP and a primary care practice administrator. There are a lot of important issues not mentioned - the background population demographics (age distribution), whether appointments are for monitoring chronic health problems or maybe pregnant women visiting midwives, advertising of extended intervals, whether text reminders are used.

R6.12 Author response: Thank you for your thoughts. The final evaluation report contains more detailed information regarding the implementation of the service, including staffing issues, services delivered, and communication approaches taken. Given word count is limited we are restricted in the depth we can go into regarding implementation and as such we reference the evaluation report for further information (it was not available prior to submission) (Reference 13). We hope the changes made in response to the earlier comments by the reviewer and the signposting to where further details are available, are sufficient to satisfy the reviewer.

Reviewer: 7

Reviewer Name: Helen Purtill

Institution and Country: University of Limerick, Ireland

Please state any competing interests or state 'None declared': None

Please leave your comments for the authors below

This study aims to examine the uptake of extended primary care service within 13 centres in Greater Manchester Area in England in 2016. The authors are to be commended for an interesting study, a well written manuscript and detailed descriptions of the data through the tables, graphs and appendices. However, I've strong concerns about the statistical models and the conclusions drawn from the analyses, as outlined below.

Abstract:

Objectives - The first objective is clear. The secondary objective looks to examine patient differences across the appointment types and to compare the demographic profile of the extended primary care service with that of core hour users.

- the first objective relates to the appointments

- the second objective relates to the patients that make the appointments

Participants - 1,261,326 are not the participants. This is the population size with access to the 5 CCGs. In this study there is a sample of 42,472 appointments. There is only have information on those who make appointments (not those who don't make the appointments). Information is not given in the study on how many participants made the 32041 appointments over the four CCGs analysed for the patient demographic analysis - it would be good to have a measure of the exact number of patients made the 32041 appointments and thus have an idea of the number of patients making repeat visits.

R7.1 Author response: We have replaced participants with patients and edited the abstract to note the focus is on extended access appointments that were available to these patients. Unfortunately we do not have the ability to identify whether there are repeat attendees in the service (this is discussed below).

In the results section of the abstract it is stated that "Uptake increased over the 2016 calendar year" - this wasn't the case. In table 3, the adjusted marginal percentages for "Appointment Used" for August is lower than July and June. The adjusted marginal percentage for December is similar to April - there is no clear pattern of an increase.

R7.2 Author response: Whilst we maintain that there appears to be an increase over time, we accept the increase over time is not consistent over the period and have mentioned this and dampened claims of a trend in increasing uptake accordingly (Page 11, paragraph 2). We have also removed this statement from the abstract.

Line 25 on Page 2: "Pre-booked appointment participants were less likely to be female" - this is from the Probit analysis presented in Table 4. But in this regression model there are a mixture of variables that measure patient characteristics (eg gender, age) and variables that measure time (day of week, month of year) and variables that measure the appointment status. If you'd like to determine whether female patients are more likely to pre-book an appointment compared to male patients, you should not control for day, or month, or appointment type - these are not measured on the patient. The analysis as presented in Table 4 (using the appointment as the unit of analysis) cannot be interpreted as if the patient was the unit of analysis.

Even if the analysis was interpreted correctly a difference of 0.66 percent points is an extremely small effect and unlikely to be meaningful in the context of the study.

R7.3 Author response: Thank you for highlighting this discrepancy. In the technical descriptions we had noted the analyses was appointment level but this was not clearly defined in the methods of the

main manuscript and, as the reviewer points out, interpretation was slightly off-track. We have rephrased the patient demographics analyses to better represent that the findings are from an appointment level analysis (Page 14, paragraph 1; Page 15, paragraphs 1 and 2). In the Methods section we have made reference to the unit of analysis and expanded the discussion of gender and age to that of the patient booking the appointment (Page 8, paragraph 1).

Line 29 & 30 of pg 2: Uptake improved over time, for all week days. This statement is too strong for the analysis in Table 3. In table 3, the adjusted marginal percentages for "Appointment Used" for August is lower than July and June. The adjusted marginal percentage for December is similar to April - there is no clear pattern of an increase.

R7.4 Author response: Please see response above (R7.2) concerning the dampening of the language used here.

Line 5 of Pg 8 "Probit models were chosen as these give the absolute variation in uptake". I understand what is meant but I'd prefer to see something like "Probit models were chosen as these estimate differences in the probability of uptake across categories of the predictor variables". The word "variation" is more usually seen in the context of quantitative outcome variables (eg The explained variation in by a fitted linear regression model).

R7.5 Author response: We have expanded the Methods section (Page 7, paragraph 1) and technical descriptions (Supplementary Text 2) to further justify the probit model with explicit reference to the alternative logistic approach. We have incorporated the reviewers helpful suggestion in these edits.

Analysis of Uptake:

It is not clear in this section if "uptake" refers to the appointment being booked or if it refers to the appointment being booked and used. Please give very specific definition for each case and use that definition throughout the analysis.

R7.6 Author response: We have now refrained from using uptake beyond the primary analyses to avoid any confusion and have made clearer where appointments booked and used and booked and not attended are the focus throughout the manuscript. We have edited the column headers in Table 1 and 3 to aid in clarity too.

1. CCG 5 is chosen as the base centre but doesn't have any weekday appointments (see Table S1) and so the model will have empty cells for day of week Mon-Fri for CCG5. A probit model should not have empty cells.

R7.7 Author response: Please note that the observations for CCG5 will exist only for days in which appointments are available, namely Saturdays. CCG5 maintains in the analysis because the outcome variable is not defined by day of week.

2. January is compared as the base month but looking at the start dates it is not surprising at all that the January numbers are so low (Table S1) - many hubs don't start until the middle, end or after January. February would seem a more reasonable comparison group for the probability of appointments booked and probability of appointment used.

R7.8 Author response: We have maintained the use of January as the base category for two reasons. First, to aid in interpretation of the results, it is likely to be simpler to understand the estimates in a chronological manner. Second, the analyses assesses rates of uptake (appointments booked and appointments used) and not volume, as such the smaller volume of appointments available in January should not bias the findings presented.

3. The title of Table 3 "Rates and probability models for appointment use" doesn't make sense. Are all the figures in table 3 estimated by the Probit models? Or are the % booked summary statistics?

R7.9 Author response: The % are summary statistics. We have expanded the column titles to reflect where probability model estimates are evident and defined the probability models more clearly in the footnotes to Table 3 and Table 4.

4. CCG is used to control for CCG area and standard errors are adjusted for patients being clustered within these areas. But there could be a hub effect. Hub could be included as a random effect in the analysis instead of the fixed effect of CCG.

R7.10 Author response: We have now clustered at the patient booking the appointments practice level which is more granular than the hub v non-hub level.

Lines 15 to 20 on page 13: The conclusions given from a non-significant result are incorrect. If a statistical test is non-significant you cannot imply there is equivalence - only that you don't have evidence in your data of the effect you were testing for. To test for an interaction of Sunday by month (ie if the effect of Sunday changed overtime) it would be sufficient to include an indicator variable "Sunday" coded as 1 = yes and 0 = no, the dummy variables indicating month, the CCG (or Hub) variables and Sunday by month interaction variables.

R7.11 Author response: Please note that Supplementary Table S5 provides the results of the model including Sunday-calendar month interactions with the method taken matching the suggestion by the reviewer. We have dampened the claim of equivalence to claims of similarity (Page 12, paragraph 1). Our inference was based on the test that the estimated effect of a variable is equal to zero, if not rejected then this implies there is no significant difference between this variable and the base category in the data, this is why we claim similarity.

Patient Demographic Analysis:

The research question surrounds the patients, and examining patient demographics associated with Nurse V GP and Pre-booked V Same day appointments. But the unit of analysis presented in the Probit regression (Table 4) is an appointment booked, not a patient. To run and interpret the analysis with respect to patient demographics the analysis needs to be run using the patient as the unit of the analysis (patient variables are gender, age and CCG or hub attended). You need to consider if there are repeat patients. If there are not many repeat patients you may assume independence of observations but this needs to be clearly stated as an assumption and you need to state it as a limitation (ie that the same person may be included in the analysis twice).

R7.12 Author response: We have rephrased a large portion of the patient demographics analyses to align with the appointment level analysis that is conducted (Page 14, paragraph 1; Page 15, paragraphs 1 and 2). We have also included a change in the sub-section title to 'Analysis of demographics of patients booking an extended access appointment'. We are unable to identify whether there are repeat patients in the data and as such we have noted this as a limitation (Page 17, paragraph 1).

An analysis of the patient demographics vs booked appointments and used appointments is interesting - from the descriptive Table S1 it appears that more females use the service compared to males, as also seen in the graph on the left of Figure 4, which shows the differences in the shape of the age profile of male and female patients. But, maybe the population is younger with more females? A descriptive table of Nurse Vs GP and Pre-booked Vs Same day across the different variables would be interesting too.

R7.13 Author response: Our primary interest here lies in the differences of the populations using the extended access service and those using core hour services. To this effect the general population of the area is not that helpful (both serve the same population). However, we do accept that as a stand alone figure, the extended access demographics should be considered alongside the population of the area. We have therefore added the demographics of the area to Figure 4. We have also added Supplementary Table S8 which contains summary statistics for the types of appointments and DNA groupings.

Would it be of interest to examine factors associated with DNA appointments (ie whether or not the appointment was used)?

Eg Are patient demographics predictive of DNA? Or you could use the appointment as the unit of analysis to examine if any appointment factors associated with DNA? A regression model could answer: are appointments made for GPs or Nurses more likely to be DNA? Are appointments made on same day v pre-booked more likely to be DNA? etc, are appointments made by a female, made on different days of the week more likely to be DNA.. etc..

R7.14 Author response: Thank you for this suggestion. Please see response to reviewer 2 (R2.8). We have now included these results in the manuscript.

Comparisons of extended access appointment to core hour users:

Yes, I'd agree that caution needs to be exercised because the age bands do not align in the datasets. You are also assuming that the gender and age profile for patients from in England (GPPS survey) is representative of the patient profile for core use in the Greater Manchester area - Is this the case? It should be included in a limitation that this may not be the case.

R7.15 Author response: Please refer to the discussion of the GPPS analysis in the methodology section, here we note "The GPPS sample was reduced to patients registered with a practice within one of the four CCG areas under evaluation..." (Page 9, paragraph 1). To be more transparent we have added further clarity in the footnote to Supplementary Table S10.

Discussion: Line 56, 57 - should be "..... who accounted for almost a quarter of all female extended access...."

R7.16 Author response: Thank you for identifying this, we have edited the sentence accordingly (Page 16, paragraph 1).

A table of descriptive statistics for GP V Nurse and Pre booked Vs Same Day across the patient demographics would be informative.

R7.17 Author response: We have included an additional Supplementary Table that provides the descriptive statistics for appointment type as suggested in an earlier comment (R7.13).

VERSION 2 – REVIEW

REVIEWER	Professor Shona Kelly Sheffield Hallam University Sheffield, UK
REVIEW RETURNED	10-Jun-2019

GENERAL COMMENTS	This is much easier to read. It is interesting to read it in the context of the other UK evaluations of PMCF as everyone had trouble getting decent data.
--

REVIEWER	Louis Levene University of Leicester
REVIEW RETURNED	12-Jun-2019

GENERAL COMMENTS	General points: 1. The authors have made huge efforts to address the (7!!) reviewers' comments/suggestions. 2. The narrative is clearly set out and the different sections of the paper are well linked.
--

	3. Apologies for some of my comments being 'new' and not stated in my previous review, but I have tried to look at this paper from a 'fresh' perspective. Specific points:  1. The 3 aims are now stated in the abstract, at the end of the introduction and in the discussion. As there are multiple aims and there is no specific hypothesis being tested, this study is exploratory and this should have been stated in the introduction. 2. The authors have rightly acknowledged that 3rd aim, comparing users of extended hours with those of core services, was not met (nor mentioned in the conclusion), due to the incompatibility of GPPS data with the study's dataset. Despite the authors' best efforts, I am not persuaded that this comparison was ever feasible in the first place. Should the authors consider dropping this aim? 3. Combining data from 4 CCG schemes that provided and delivered services in very different ways has caused problems. The authors have tried to address this, but it would be helpful for a statistician to advise if their methods were optimal and have been successful. I think that the combination of probably quite heterogeneous user populations is responsible for some quite large 95% CI in tables 3 & 4. 4. I note that the authors only used 2 variables, age and gender to describe these populations, and also their reasons for not including deprivation. Yes, their 2nd aim has been addressed; however, the lack of any other variables, especially relating to the socio-economic or the health status of the service users, is a limitation and prevents a more in depth exploration of the implications, as these variables may have given further insight into who is more likely to use non-core hours services in primary care. Age/gender bands are likely to be heterogeneous. Are the data available to include further variables in the analyses? 5. Despite the CCG populations being roughly similar in size, more than 70% of the appointments analysed were in 1 CCG only. Furthermore, due to incomplete data, appointments in one (the largest) CCG scheme were excluded. The one CCG is thus over-represented in the dataset, compared to the remaining CCGs. It is unclear from the text whether the authors were concerned about this and if so how they addressed it. These are potentially important limitations. 6. I could not find a dedicated subsection of conclusions in the discussion section of the main text to match those stated in the abstract.
--	---

REVIEWER	Sam Watson University of Warwick, UK
REVIEW RETURNED	20-Jun-2019

GENERAL COMMENTS	The authors have addressed my previous comments and the article is clear and will be a useful addition to the literature. I have only one minor comment that may be useful to the authors for clarity: - page 7, lines 27-30: Probit regression coefficients cannot be interpreted as absolute risk differences, although they could be derived from an estimated probit model. However they could also be derived from a logit model in the same way, e.g. difference in predicted outcomes or average marginal effects, for example. As such I don't see this as a justification for probit over logit link function (which is not to say logit should be used here).
---

REVIEWER	John Ford Cambridgeshire and Peterborough CCG
REVIEW RETURNED	20-Jun-2019

GENERAL COMMENTS	I'm happy with the revised manuscript
---------------------------------------

REVIEWER	Irene Stratton Gloucester Hospitals NHS FT UK
REVIEW RETURNED	24-Jun-2019

GENERAL COMMENTS	There are some remaining glitches - in the text reporting percentages to no decimal or just one decimal place would be sufficient - there seems to be variation between no numbers after the decimal point and 2.... as here: "we found age group 20-29 to be the most prevalent group of patients using the service (16.96%) and a higher proportion of patients from the 20-69 age group (73.09% compared to 57% aged 20-65)" Perhaps you also mean 'largest' rather than 'most prevalent'? Some of the sentences are very long and don't always make sense: Evaluations of patient demographics using extended access appointments for urgent care in Sheffield, UK, found users were predominantly female (60.0%) and aged under 60 (85.5%), with 19.05% aged under 5 and 29.35% aged under 16 and whilst ethnicity data had a high proportion of missing data, deprivation was captured finding users from deprived areas dominated use.[12]" Here again the number of decimal places bounces around.....
--

REVIEWER	Helen Purtill University of Limerick, Ireland
REVIEW RETURNED	17-Jun-2019

GENERAL COMMENTS	The authors have addressed my previous concerns on the analysis and presentation of results. I just have a couple of comments:  1. In Supplementary text 2 the sentence "Regression models were estimated because there are instances of missing data meaning the dataset is not a complete picture of activity in the population." The regression models used in the analysis don't impute data for any missing values. They provide a complete case analysis of the data. I would there recommend removing this sentence. 2. Also remove the line "Regression models were estimated because there are instances of missing data meaning the dataset is not a complete picture of activity in the population." from page 18/19 of the main paper. 3. Line 7 page 8 - should be "three" instead of "two"
---

VERSION 2 – AUTHOR RESPONSE

Reviewer: 1

Reviewer Name: Professor Shona Kelly

Institution and Country: Sheffield Hallam University, Sheffield, UK

Please state any competing interests or state 'None declared': none declared

Please leave your comments for the authors below

This Is much easier to read.

It is interesting to read it in the context of the other UK evaluations of PMCF as everyone had trouble getting decent data.

Author response: Thank you for your comments. We are pleased the revised manuscript is clearer. We agree too that the study gives a useful insight in the extended access service, particularly given the scarcity of data currently surrounding the initiative.

Reviewer: 2

Reviewer Name: Louis Levene

Institution and Country: University of Leicester

Please state any competing interests or state 'None declared': None declared.

Please leave your comments for the authors below

General points:

1. The authors have made huge efforts to address the (7!!) reviewers' comments/suggestions.
2. The narrative is clearly set out and the different sections of the paper are well linked.
3. Apologies for some of my comments being 'new' and not stated in my previous review, but I have tried to look at this paper from a 'fresh' perspective.

Author response: Thank you for your comments. We are pleased the revised manuscript has a better flow. Thank you also for your additional comments below.

Specific points:

1. The 3 aims are now stated in the abstract, at the end of the introduction and in the discussion. As there are multiple aims and there is no specific hypothesis being tested, this study is exploratory and this should have been stated in the introduction.

Author response: We have added '...from these exploratory analyses...' to page 5, paragraph 2 to highlight there are no formal hypotheses tests conducted.

2. The authors have rightly acknowledged that 3rd aim, comparing users of extended hours with those of core services, was not met (nor mentioned in the conclusion), due to the incompatibility of GPPS data with the study's dataset. Despite the authors' best efforts, I am not persuaded that this comparison was ever feasible in the first place. Should the authors consider dropping this aim?

Author response: We feel the GPPS comparison is the best dataset available for understanding the demographics of patients using general practice in England (as we note on Page 8, paragraph 4). We appreciate and acknowledge the flaws of the comparisons of patients in the two datasets but feel there is value in the analyses. In addition to the recognition that this was exploratory (see Reviewer 2's previous comment), we have added '...making it not possible to draw direct comparisons of age groups' to the discussion to highlight further that comparisons were not exact (Page 17, paragraph 1). We hope the changes reflect more clearly the limitation of the approach taken.

3. Combining data from 4 CCG schemes that provided and delivered services in very different ways has caused problems. The authors have tried to address this, but it would be helpful for a statistician to advise if their methods were optimal and have been successful. I think that the combination of probably quite heterogeneous user populations is responsible for some quite large 95% CI in tables 3 & 4.

Author response: Please note that to account for variations in scheme delivery and populations we have adjusted for CCG scheme in the regression analyses and also cluster the standard errors by CCG. The first approach enables the estimated effects to take into account differences across the CCG schemes and populations. The second approach ensures that statistical inference acknowledges that there is commonality within CCG for appointments and variation across CCGs. Please note that we provide details regarding this on Page 7, paragraph 3.

4. I note that the authors only used 2 variables, age and gender to describe these populations, and also their reasons for not including deprivation. Yes, their 2nd aim has been addressed; however, the lack of any other variables, especially relating to the socio-economic or the health status of the service users, is a limitation and prevents a more in depth exploration of the implications, as these variables may have given further insight into who is more likely to use non-core hours services in primary care. Age/gender bands are likely to be heterogeneous. Are the data available to include further variables in the analyses?

Author response: Unfortunately we are limited in the data that can be explored and have utilised all available to us. Please note the data is sourced from a minimum dataset request made by the authors to the service provider. We detail the data collected and the reasons further data was not available on Page 6, paragraph 1. We have added an additional sentence to highlight that we were unable to obtain data on health status or outcomes of patients attending the service ('We were unable to obtain health status or outcomes of patients attending the service': Page 6, paragraph 1).

5. Despite the CCG populations being roughly similar in size, more than 70% of the appointments analysed were in 1 CCG only. Furthermore, due to incomplete data, appointments in one (the largest) CCG scheme were excluded. The one CCG is thus over-represented in the dataset, compared to the remaining CCGs. It is unclear from the text whether the authors were concerned about this and if so how they addressed it. These are potentially important limitations.

Author response: The reviewer makes a valid point. CCG 2 is of a similar size to the other CCGs yet provided the vast majority of appointments (Table 1). Please note that this is one of the reasons for including CCG adjustment in the analyses and clustering the standard errors at the CCG level (see also Reviewer 2's third comment).

6. I could not find a dedicated subsection of conclusions in the discussion section of the main text to match those stated in the abstract.

Author response: The Discussion section of the manuscript contains a more detailed discussion of the key points highlighted in the abstract which is to an extent constrained to follow a certain structure. For example, we have refrained from mentioning implications in the principal findings section of the discussion yet in the abstract we merge findings with implications due to word count.

Reviewer: 7

Reviewer Name: Helen Purtill

Institution and Country: University of Limerick, Ireland

Please state any competing interests or state 'None declared': None

Please leave your comments for the authors below

The authors have addressed my previous concerns on the analysis and presentation of results.

Author response: Thank you for your previous comments and we are pleased to have sufficiently addressed these.

I just have a couple of comments:

1. In Supplementary text 2 the sentence "Regression models were estimated because there are instances of missing data meaning the dataset

is not a complete picture of activity in the population."

The regression models used in the analysis don't impute data for any missing values. They provide a complete case analysis of the data. I would there recommend removing this sentence.

Author response: Thank you for highlighting this. On reflection we agree with the reviewer that this was not clearly detailed. We have removed the sentence as suggested (Supplementary Text S2, paragraph 3).

2. Also remove the line "Regression models were estimated because there are instances of missing data meaning the dataset is not a complete picture of activity in the population." from page 18/19 of the main paper.

Author response: In line with the previous comment we have removed the sentence as suggested (Page 7, paragraph 2).

3. Line 7 page 8 - should be "three" instead of "two"

Author response: Thank you for spotting this, we have changed as suggested (Page 8, paragraph 1).

Reviewer: 4

Reviewer Name: John Ford

Institution and Country: Cambridgeshire and Peterborough CCG

Please state any competing interests or state 'None declared': None

Please leave your comments for the authors below

I'm happy with the revised manuscript

Author response: Thank you for your previous comments and we are pleased to have sufficiently addressed these.

Reviewer: 3

Reviewer Name: Sam Watson

Institution and Country: University of Warwick, UK

Please state any competing interests or state 'None declared': None declared

Please leave your comments for the authors below

The authors have addressed my previous comments and the article is clear and will be a useful addition to the literature. I have only one minor comment that may be useful to the authors for clarity:

- page 7, lines 27-30: Probit regression coefficients cannot be interpreted as absolute risk differences, although they could be derived from an estimated probit model. However they could also be derived from a logit model in the same way, e.g. difference in predicted outcomes or average marginal effects, for example. As such I don't see this as a justification for probit over logit link function (which is not to say logit should be used here).

Author response: Thank you for your previous comments and we are pleased to have addressed these and to see agreement of the value of the manuscript to the literature. The reviewer is correct, as such we have removed the sentence from the manuscript (Page 7, paragraph 2) and Supplementary Text S2. Given the text was seeking to justify the use of absolute risk differences over odds-ratios we believe the removal of the sentence does not impair the manuscript.

Reviewer: 6

Reviewer Name: Irene Stratton

Institution and Country: Gloucester Hospitals NHS FT, UK

Please state any competing interests or state 'None declared': None

Please leave your comments for the authors below

There are some remaining glitches - in the text reporting percentages to no decimal or just one decimal place would be sufficient - there seems to be variation between no numbers after the decimal point and 2.... as here:

"we found age group 20-29 to be the most prevalent group of patients using the service (16.96%) and a higher proportion of patients from the 20-69 age group (73.09% compared to 57% aged 20-65)"

Author response: Please note that the inconsistency in decimal places is derived from variation in the use of decimal places across the studies being referenced. Reference 11 did not use decimal points, we have amended the comparisons to this study to ensure all comparisons are to no decimal places (Page 19, paragraph 2). Reference 12 reported to one decimal point, we have amended the figures for reference 12 in light of a minor error in that manuscript (13.15 reported rather than 13.1% in Table 1 of Reference 12).

Perhaps you also mean 'largest' rather than 'most prevalent'?

Author response: We have amended the text as suggested by the reviewer (Page 19, paragraph 2).

Some of the sentences are very long and don't always make sense:

Evaluations of patient demographics using extended access appointments for urgent care in Sheffield, UK, found users were predominantly female (60.0%) and aged under 60 (85.5%), with 19.05% aged under 5 and 29.35% aged under 16 and whilst ethnicity data had a high proportion of missing data, deprivation was

captured finding users from deprived areas dominated use.[12]"

Here again the number of decimal places bounces around.....

Author response: Thank you for highlighting this. We have amended the sentence to make the findings from this study clearer (Page 19, paragraph 2).

VERSION 3 – REVIEW

REVIEWER	Louis Levene University of Leicester, United Kingdom
REVIEW RETURNED	31-Jul-2019

GENERAL COMMENTS	Thank you for responding to my comments. I am happier with the paper now. Although how extended access is funded and organised is changing and these will have a bearing on future research, there are important messages in the paper that are still very relevant.
--